

# Interannual to decadal sea level variability in the subpolar North Atlantic: The role of propagating signals

Denis L. Volkov[1,2], Claudia Schmid[2], Leah Chomiak[3], Cyril Germineaud[4], Shenfu Dong[2], Marlos Goes[1,2]

[1]Cooperative Institute for Marine and Atmospheric Studies, University of Miami, Miami, Florida 33149, USA
[2]NOAA Atlantic Oceanographic and Meteorological Laboratory, Miami, Florida 33149, USA
[3]Rosenstiel School of Marine and Atmospheric Science, University of Miami, Miami, Florida 33149, USA
[4]Mercator Ocean International, Toulouse, France

*Correspondence to*: Denis L. Volkov (denis.volkov@noaa.gov)

**Abstract.** The gyre-scale, dynamic sea surface height (SSH) variability signifies the spatial redistribution of heat and
freshwater in the ocean, influencing the ocean circulation, weather, climate, sea level, and ecosystems. It is known that the
first empirical orthogonal function (EOF) mode of the interannual SSH variability in the North Atlantic exhibits a tripole gyre
pattern, with the subtropical gyre varying out-of-phase with both the subpolar gyre and the tropics, influenced by the low-
frequency North Atlantic Oscillation. Here, we show that the first EOF mode explains the majority (60-90%) of the interannual
SSH variance in the Labrador and Irminger Seas, whereas the second EOF mode is more influential in the northeastern part of
the subpolar North Atlantic (SPNA), explaining up to 60-80% of the regional interannual SSH variability. We find that the
two leading modes do not represent physically independent phenomena. On the contrary, they evolve as a quadrature pair
associated with a propagation of SSH anomalies from the eastern to the western SPNA. This is confirmed by the complex EOF
analysis, which can detect propagating (as opposed to stationary) signals. The analysis shows that it takes about 2 years for
sea level signals to propagate from the Iceland Basin to the Labrador Sea, and it takes 7-10 years for the entire cycle of the
North Atlantic SSH tripole to complete. The observed westward propagation of SSH anomalies is linked to shifting wind stress
curl patterns and to the cyclonic pattern of the mean ocean circulation in the SPNA. The analysis of regional surface buoyancy
fluxes in combination with the upper-ocean temperature and salinity changes suggests a time-dependent dominance of either
air-sea heat fluxes or advection in driving the observed SSH tendencies, while the contribution of surface freshwater fluxes
(precipitation and evaporation) is negligible. We demonstrate that the most recent cooling and freshening observed in the
SPNA since about 2010 was mostly driven by advection associated with the North Atlantic Current. The results of this study
indicate that signal propagation is an important component of the North Atlantic SSH tripole, as it applies to the SPNA.

## 1 Introduction

Ocean and atmosphere dynamics induce regional sea level changes with amplitudes that are often several times greater than
the global mean sea level rise of about 3.5 mm yr[-1] (Meyssignac & Cazenave, 2012; Cazenave et al., 2014; Volkov et al., 2017;
Chafik et al., 2019). On the timescales from seasonal to longer, this dynamic sea level variability (after the global mean sea



level has been subtracted) is mainly due to changes in the density of water column, driven by surface buoyancy fluxes and the advection of heat and freshwater by ocean currents (Gill & Niiler, 1973; Ferry et al., 2000; Volkov & van Aken, 2003; Cabanes et al., 2006). The associated changes in the temperature and salinity of the water column determine the thermosteric and halosteric components of sea level variability, respectively, which often offset each other. Away from polar regions, the

dynamic sea level variability generally follows the thermosteric changes and, therefore, represents a good proxy for the upper-ocean heat content variability (e.g., Chambers et al., 1998; Volkov et al., 2019a, 2020).

It has been shown that on interannual to decadal timescales, the variability of sea surface temperatures (SST), sea surface height (SSH), and oceanic heat content in the North Atlantic exhibits a tripole pattern with an out-of-phase relationship between the subtropical gyre and both the subpolar gyre and the tropics (Tourre et al., 1999; Watanabe & Kimoto, 2000; Marshall et

al., 2001; Häkkinen, 2001; Volkov et al., 2019a, 2019b). The tripole is correlated with the low-frequency North Atlantic Oscillation (NAO; Marshall et al., 2001; Volkov et al., 2019b), which is the leading mode of atmospheric variability in the North Atlantic (Hurrell et al., 2003). Earlier studies suggested that both the SST tripole and the SSH tripole represent the ocean's response to the atmosphere-ocean heat flux (Cayan, 1992; Häkkinen, 2001). However, more recent research demonstrated that the upper ocean heat content and, consequently, SSH change in the North Atlantic is dominated by the

oceanic heat transport divergence (Häkkinen et al., 2015; Piecuch et al., 2017). The latter is modulated by the Atlantic meridional overturning circulation (AMOC). For example, an abrupt reduction in the AMOC observed at 26.5ºN in 2008-2009 led to a cold SST anomaly and a decrease of the thermosteric sea level (and, hence, the upper ocean heat content) in the entire subtropical gyre in 2009-2010 (Josey et al., 2018; Volkov et al., 2019a). Using observations and an ocean model, Volkov et al. (2019b) showed that the tripole-related SSH variability between 26ºN and 45ºN (the subtropical band of the tripole) is

correlated with the AMOC and meridional heat transport at 26.5ºN, and with the low-frequency NAO.

In this study, we focus on the subpolar North Atlantic (SPNA; Figure 1), which is one of the key regions for deep water formation and, as such, it plays an important role in driving the AMOC. Warming, salinification, spindown, and contraction of the North Atlantic subpolar gyre were observed in the 1990s to mid-2000s (Häkkinen and Rhines, 2004; Sarafanov et al., 2008; Holliday et al., 2008). This warming was associated with the positive phase of the Atlantic Multidecadal Oscillation

(e.g., Enfield et al., 2001), partly driven by the strengthening of the North Atlantic Current (NAC) and the AMOC (Msadek et al., 2014). A cooling tendency in the SPNA started in 2006, and it ended with an exceptionally cold anomaly in 2015, termed the 'cold blob' (Ruiz-Barradas et al., 2018; Chafik et al., 2019). Several studies have attributed the 'cold blob' observed in the SPNA to the dramatic wintertime heat loss event in 2014–15 (Josey et al., 2018; Grist et al., 2016; Duchez et al., 2016). Bryden et al. (2020), however, suggested that the 'cold blob' could also be linked to the 2008-2009 reduction of the AMOC observed

at 26.5ºN. Piecuch et al. (2017) showed, however, that horizontal gyre circulations provide a greater contribution to heat divergences in the SPNA than vertical overturning circulations.

Along with the cooling anomaly, a slowdown of the warm and salty NAC and redistribution of the fresher Arctic waters led to an unprecedented freshening of the SPNA in 2012-2016 (Holliday et al., 2020) that rapidly communicated with the deep



waters (Chafik and Holliday, 2022). The 2013-2015 cold event in the SPNA also intensified deep ocean convection in the
Irminger and Labrador basins (de Jong & de Steur, 2016; Yashayaev & Loder, 2016). While the reduced AMOC may have
influenced the intense cooling in the subpolar gyre around 2015, the resulting intensification of deep convection is expected
to eventually increase the strength of the AMOC (Frajka-Williams et al., 2017). A recent analysis showed that the 2006-2015
cooling trend in the SPNA may have reversed in 2016 with a large-scale warming in the central and eastern parts of the domain
due to an enhanced advection of warm and saline waters from the subtropical gyre (Desbruyères et al., 2021).

Since large-scale ocean circulation is an important driver for the observed low-frequency changes in the North Atlantic,
propagation of signals across the basin and between the subtropical and subpolar gyres is an intrinsic element of the variability.
While westward propagating Rossby waves and eddies provide an effective mechanism for energy transfer in the tropics and
within the subtropical gyres, advection by ocean currents plays a primary role for the meridional and zonal transports along
western boundary currents and at high latitudes. For example, Fu (2004) identified a propagation of the interannual sea level
signal from the subtropical region to the eastern end of the Gulf Stream extension, demonstrating that the SSH variability is
not just a steric response to the heat flux forcing, but also involves a dynamic response. Using a simple thermodynamic model,
Dong and Kelly (2004) showed that advection by geostrophic currents can largely contribute to interannual variations of the
upper ocean heat content in the Gulf Stream region. Desbruyères et al. (2021) hypothesized that the warming signal observed
in the eastern SPNA since 2016 would progressively propagate westward into the Irminger and Labrador Basins.

In areas where propagation is important, analyses based on the widely used Empirical orthogonal function (EOF)
decomposition may be misleading, because each individual EOF pattern represents a standing oscillation. Propagating signals
are often represented as a linear combination of more than one EOF pattern, such as a pair, with one mode leading the other
by 90° of phase in space and time (e.g., Roundy, 2015). In this case, treating consecutive EOF modes as independent physical
processes is not appropriate. As was shown previously, the North Atlantic SSH tripole is the leading EOF mode of the
interannual SSH variability in the region (Figure 2; Volkov et al., 2019b). The time evolution of the tripole suggests that the
subpolar (subtropical) gyre was experiencing warming (cooling) in 1993-2010 and cooling (warming) in 2011-2016, and it
was still in a cold (warm) phase in 2020 (Figures 2 and 3). However, these tendencies differ from those reported in recent
studies showing that a cooling in the SPNA started in 2006 and was followed by a warming since 2016 (Chafik et al., 2019;
Desbruyères et al., 2021).

In this paper, we show that this discrepancy is due to a signal propagation, which is not reflected by the present definition of
the North Atlantic SSH tripole. The main objective of this study is to revisit the definition of the North Atlantic SSH tripole
by accounting for signal propagation and explore the role of the tripole in the SPNA. Nearly 30 years of quasi-global, regular
satellite altimetry measurements allow us not only to resolve the interannual variability, but also gain a preliminary insight
into decadal changes. Therefore, we specifically focus on interannual to decadal time scales. Furthermore, we explore how the
leading modes of the variability in the SPNA are related to atmospheric wind and buoyancy forcing and to advection by ocean
currents.



## 2 Data

### 2.1 Satellite altimetry measurements

Satellite altimetry has provided accurate, nearly global, and sustained observations of sea level since the launch of the
Topex/Poseidon mission in August 1992. We use the monthly and daily maps of SSH anomalies for the time period from
January 1993 to December 2020 processed and distributed by the Copernicus Marine and Environment Monitoring Service
(CMEMS; http://marine.copernicus.eu). The daily maps are used only for the computation of eddy propagation velocities
(Section 3.3). The maps are produced by the optimal interpolation of measurements from all the altimeter missions available
at a given time. Prior to mapping, the along-track altimetry records are routinely corrected for instrumental noise, orbit
determination error, atmospheric refraction, sea state bias, static and dynamic atmospheric pressure effects, and tides (Pujol et
al. 2016). The SSH anomalies produced by CMEMS are computed with respect to a twenty-year (1993-2012) mean sea surface,
but we center them around the record-long mean. We also subtract the global mean sea level from SSH anomaly time series at
each grid point to focus on local dynamic sea level variability not related to global changes.

### 2.2 Hydrographic data

To relate sea level variability to subsurface processes, we use the EN4 monthly gridded profiles of temperature and salinity
(version EN.4.2.2 with mechanical/expendable bathythermograph (MBT/XBT) profile data bias adjustments from Gouretski
and Reseghetti, 2010) for the period from January 1993 to December 2020 (Good et al., 2013). The profiles are based on the
objective analysis of hydrographic observations (e.g., XBT, MBT, bottle, CTD, and Argo). The number of Argo profiling
floats in the ocean was growing from a very sparse array of 1000 profiling floats in 2004 to a global array of more than 3000
instruments from late 2007 to the present. This means that maps in 2004-2007 and especially before 2004 are less accurate
than maps in 2008-2020. The EN4 temperature and salinity profiles are used to calculate the steric SSH anomalies ($SSH_{ST}$;
due to changes in density) by integrating in-situ density anomalies with respect to the time mean and over the upper 1,000 m,
which is the depth interval occupied by the NAC. The thermosteric ($SSH_T$; due to temperature changes only) and halosteric
($SSH_S$; due to salinity changes only) contributions to SSH are calculated by integrating in-situ density anomalies with respect
to the time mean values of salinity and temperature, respectively. The obtained $SSH_T$ and $SSH_S$ are proportional to the upper
1,000 m heat and freshwater contents, respectively. Similar to the processing of altimetry data, the global mean $SSH_{ST}$, $SSH_T$,
and $SSH_S$ are subtracted from the respective time series at each grid point. To analyze the role of the mean ocean circulation,
the individual Argo temperature and salinity profiles and float trajectories from the U.S. Argo Data Assembly Center
(https://www.aoml.noaa.gov/argo/#argodata) are used to compute the time-mean adjusted geostrophic velocities at 1000-dbar
(Schmid, 2014; see Section 3.3).



## 2.3 Atmospheric data

The observed changes in SSH are analyzed jointly with atmospheric forcing fields provided by ERA5 climate reanalysis produced by the European Centre for Medium-Range Weather Forecasts (ECMWF) and distributed by Copernicus (https://climate.copernicus.eu/climate-reanalysis). Specifically, we use the monthly averaged fields of sea level pressure (SLP),

10-m wind velocities, surface wind stress, and surface heat (shortwave and thermal radiation, the sensible and latent heat fluxes) and freshwater (precipitation, evaporation) fluxes from January 1979 to December 2020 (Hersbach et al., 2019). In addition, we also use the monthly station-based NAO index, based on the normalized SLP difference between Lisbon (Portugal) and Reykjavik (Iceland) from the Climate Analysis Section of the National Center for Atmospheric Research (Hurrell et al., 2003; https://climatedataguide.ucar.edu).

**3 Methods**

To focus on interannual and longer time scales, the seasonal cycle is computed by fitting both the annual and semi-annual harmonics in a least squares sense and subtracted from all data fields and time series. The time series are further low-pass filtered with a "Lowess" filter with a cutoff period of 18 months.

### 3.1 EOF analysis

Empirical orthogonal function (EOF) analysis is widely used to reduce the dimensionality of datasets and to extract the leading (most influential) modes of variability. Individual EOF modes, if found meaningful, are often assumed to describe different physical processes. Here, we use the conventional EOF analysis (Navarra & Simoncini, 2010) to identify the leading stationary modes of the low-frequency SSH variability in the North Atlantic. The EOF analysis is applied to both the satellite altimetry (SSH) and the EN4-derived ($SSH_{ST}$, $SSH_T$, and $SSH_S$) data. Because of the scarcity of in situ observations prior to the advent

of Argo, only the EN4 data starting from 2004 was used for the EOF analysis.

For each mode $j$, the spatial pattern (map) is represented as a regression map ($EOF_j$) obtained by projecting SSH data onto the standardized (divided by standard deviation) principal component ($PC_j$) time series. Thus, the regression coefficients are in centimeters (local change of sea level) per 1 standard deviation change of the associated $PC_j$. The SSH fields can be reconstructed using a limited number of modes:

$$SSH_R(x,t) = \sum_{j=1}^{N} PC_j(t)EOF_j(x,t) \tag{1}$$

where $x$ is spatial location, and $t$ is time. The portion of local variance explained by $SSH_R$ (the selected number of EOF modes) is then estimated as

$$\sigma_j^2(x) = 100\% \times \left[1 - \frac{\mathrm{var}(SSH - SSH_R)}{\mathrm{var}(SSH)}\right] \tag{2}.$$



The conventional EOF analysis is not an effective method to identify propagating modes of variability. Therefore, the detection
of propagating modes of the low-frequency SSH variability in the North Atlantic is carried out using the Complex EOF (CEOF)
analysis, which yields the spatial and temporal amplitude and phase information (Navarra & Simoncini, 2010). In essence, the
CEOF analysis is based on the notion that a propagating signal should contain signals that are orthogonal to each other. In
practice, the CEOF analysis of a data field is similar to the conventional EOF analysis, but applied to the field augmented in a
manner such that propagating signals within it may be detected (Navarra & Simoncini, 2010). The augmentation of the data
field is achieved through forming complex time series:

$$SSH(x,t) = SSH(x,t) + i \cdot SSH^T(x,t) \tag{3}$$

where the real part is simply the original data field and the imaginary part is its Hilbert transform, representing a filtering
operation upon SSH($x,t$) in which the amplitude of each spectral component is unchanged but each component's phase is
advanced by $\pi/2$ (Horel, 1984). Unlike the conventional EOF analysis, the CEOF analysis yields complex eigenvectors for the
spatial patterns (maps) and for their principal components (time evolution). For simplicity, henceforth, we refer to the spatial
patterns as CEOFs (CEOF$_j$) and to the complex principal components as CPCs (CPC$_j$). Similar to the conventional EOF
analysis, the real and imaginary parts of CEOFs are presented by regressing SSH onto the standardized real and imaginary
parts of CPCs. The time-space progression of the CEOF$_j$ mode is obtained by multiplying CEOF$_j$(x,y) and CPC$_j$(t) by a rotation
matrix whose argument may vary from 0–360°. The real and imaginary parts of CEOFs and CPCs are also used to obtain the
spatial ($\Phi$) and temporal ($\theta$) phases of the CEOFs:

$$\Phi_j(x) = \tan^{-1} \frac{\text{Imag}(CEOF_j)}{\text{Real}(CEOF_j)}$$

$$\theta_j(x) = \tan^{-1} \frac{\text{Imag}(CPC_j)}{\text{Real}(CPC_j)} \tag{4}.$$

**3.2 Forcing mechanisms**

The statistical relationship between SSH and atmospheric circulation is established by regressing SLP, 10-m wind velocity,
and surface wind stress fields on PC$_j$ and CPC$_j$. It is reasonable to assume that on interannual-to-decadal time scales the SSH
changes are mostly steric (e.g., Volkov and van Aken, 2003) and, therefore, the mass-related changes can be neglected. The
variations of SSH$_{ST}$ are related to heat and freshwater fluxes as follows (e.g., Cabanes et al., 2006):

$$\partial_t SSH_{ST} = \frac{\alpha}{\rho_0 c_p} [Q_{NET}(t) - \bar{Q}_{NET}] + \beta S_a [F_{fw}(t) - \bar{F}_{fw}] + Adv \tag{5}$$

where $Q_{NET} = Q_{SR} + Q_{TR} + Q_{SH} + Q_{LH}$ is the net surface heat flux (positive into the ocean) equal to the sum of shortwave
radiation (Q$_{SR}$), thermal (longwave) radiation (Q$_{TR}$), the sensible heat flux (Q$_{SH}$), and the latent heat flux (Q$_{LH}$); $F_{fw} = P - E$
is the surface freshwater flux (positive into the ocean) equal to the sum of precipitation (P) and evaporation (E); $\alpha$ and $\beta$ are
the coefficients of thermal expansion and haline contraction averaged over the upper 1000 m, respectively; $\rho_0$ is the reference
density; $c_p$ is the specific heat of seawater; $S_a$ is the salinity anomaly relative to a multi-year (2004-2020) mean value averaged



over the upper 1000 m; *ADV* denotes the $SSH_{ST}$ change due to the advection of density anomalies; and the overbar indicates
the climatological averages over the entire ERA5 record (1979-2020).

### 3.3 Ocean circulation and eddy propagation

The potential impact of oceanic advection in the SPNA is qualitatively analysed using (i) 1000-dbar velocities obtained from
Argo data and (ii) eddy propagation velocities estimated from daily satellite altimetry maps. The time-mean gridded adjusted
geostrophic velocities at 1000-dbar are derived from Argo and altimetry using the climatological velocity field from Argo
trajectories as the reference velocity following the methodology of Schmid (2014). This method uses Argo dynamic height
profiles and SSH from altimetry to derive synthetic dynamic height profiles on a 0.5° grid. These profiles are then used to
derive the horizontal geostrophic velocity, followed by the barotropic adjustment.

The eddy propagation velocities are calculated using a space-time lagged correlation analysis of daily SSH fields as described
in Volkov et al. (2013) and Fu (2006). A velocity vector at a particular grid point is obtained as follows. First, the correlations
between SSH anomalies at this grid point and at neighboring grid points are computed at various time lags. Second, the location
of the maximum correlation is recorded, and a velocity vector is computed using the distance between the two grid points and
the time lag. Finally, an average velocity vector weighted by correlation coefficients is computed from velocity vectors at
various time lags. To obtain the results characteristic for eddy temporal and spatial scales, the time lags were limited to less
than 70 days and the horizontal dimensions of the area for computing the correlations between the neighboring grid points
were set to about 200 km. Because SSH is an integral quantity characteristic for the full-depth thermohaline properties, the
obtained velocities are representative of property transports.

## 4 Results

### 4.1 North Atlantic SSH tripole

The North Atlantic SSH tripole is defined as the leading EOF mode ($EOF_1$) of the low-pass filtered SSH (Volkov et al., 2019b).
It is characterized by the subtropical band (~20°-45°N) varying out-of-phase with both the tropical North Atlantic (south of
~20°N) and the subpolar gyre (~45°-65°N) (Figure 2a). The tripole mode explains 27.2% of the interannual to decadal SSH
variability in the North Atlantic. The time evolution of the tripole is shown by $PC_1$ (solid blue curve in Figure 3). In 1993-
2010, there were general tendencies for the reduction of sea level in the subtropical gyre, including the Gulf of Mexico and the
Caribbean, and the increase of sea level in both the subpolar gyre and in the tropics. The rising sea levels in the SPNA were
associated with a weakening of the subpolar gyre circulation (Häkkinen and Rhines, 2004). In 2011-2015, these tendencies
reversed abruptly. Driven in part by the oceanic heat convergence, the subtropical gyre warmed considerably, which led to an
accelerated sea level rise along the U. S. southeast coast with the rates of up to five times the global mean (Domingues et al.,





2018; Volkov et al., 2019b). Opposite to the subtropical warming, a strong cooling was observed in the subpolar North Atlantic in 2013-2015 (Chafik et al., 2019).

According to EOF$_1$ (Figure 2a) and PC$_1$ (blue curve in Figure 3), the subtropical and subpolar gyres have been in warm and cold states, respectively, since 2015. However, Desbruyères et al. (2021) reported an upper ocean warming in the eastern SPNA that started in 2015. While this warming is apparently not part of the leading EOF mode, it appears to be consistent with the time evolution (PC$_2$; red curve in Figure 3) of the second EOF mode (EOF$_2$; Figure 2b). The EOF$_2$ mode explains 13.4% of the variance and exhibits a pattern that suggests changes in the strength and likely meridional shifts of the Gulf

Stream to the east of Cape Hatteras. This mode is also responsible for decadal changes in the eastern and northeastern SPNA: a generally positive SSH tendency in 1993-2003, a SSH decrease in 2004-2014, and a SSH increase since 2015 (red curve in Figure 3), consistent with the observations of Desbruyères et al. (2021).

Sea level changes associated with EOF$_1$ and EOF$_2$ are mostly steric, i.e., determined by density variations. This is confirmed by the spatial structures (Figure 2 c,d) and temporal evolutions (PCs; dotted curves in Figure 3) of EOF$_1$ and EOF$_2$ of the low-

pass filtered SSH$_{ST}$ accounting for 45.2% and 16.8% of the variance, respectively. These two leading modes are similar to those of the low-pass filtered SSH (Figure 2 a,b; solid curves in Figure 3). The correlation between PC$_1$ of SSH and PC$_1$ of SSH$_{ST}$ is 0.94, and the correlation between PC$_2$ of SSH and PC$_2$ of SSH$_{ST}$ is 0.88, significant at 95% confidence. Furthermore, the steric sea level changes are governed by changes in the upper 1000-m ocean heat content. The spatial patterns and the signs of EOF$_1$ and EOF$_2$ of the low-pass filtered SSH$_{ST}$ are determined by the thermosteric sea level variability; the EOF$_1$ and EOF$_2$

of the low-pass filtered SSH$_T$ are nearly identical to the EOF$_1$ and EOF$_2$ of the low-pass filtered SSH$_{ST}$ (compare Figures 4 a,b and 2 c,d). The correlation between the PC$_1$ of SSH$_{ST}$ and SSH$_T$ is 0.99, and the correlation between the PC$_2$ of SSH$_{ST}$ and SSH$_T$ is 0.97 (Figure 4c). The thermosteric sea level changes are partly compensated by the halosteric sea level changes, because the warmer (colder) water is usually saltier (fresher). Indeed, the EOF$_1$/PC$_1$ and EOF$_2$/PC$_2$ of SSH$_S$ (Figure 4 d-f) generally offset those of SSH$_T$ (Figure 4 a-c). For example, the cooling tendencies observed in the SPNA in 2006-2016 depicted

by EOF$_1$ and in 2004-2011 depicted by EOF$_2$ were associated with respective upper-ocean freshening.

The percentage of the local SSH variance explained by the two leading EOF modes in the SPNA exhibits the following patterns. The North Atlantic SSH tripole, characterized by the EOF$_1$/PC$_1$ of the low-pass filtered SSH, explains the majority (60-90%) of the interannual SSH variance in the Labrador Sea and in the Irminger Basins (western SPNA; Figure 5a). The EOF$_2$ mode explains a substantial amount of the interannual SSH variance (60-80%) in the northeastern SPNA, east of Greenland (Figure

5b). It is interesting to note that while the EOF$_1$ depicts the interannual SSH variability over the deep parts of the Labrador Sea and Irminger Basin, the EOF$_2$ depicts the interannual SSH variability over the shallower waters of the Rockall Plateau, parts of European continental shelf and slope, Reykjanes Ridge, Iceland shelf, and along the Irminger and East Greenland Currents. The two leading modes together (EOF$_1$ and EOF$_2$; Figure 5c) explain most of the interannual SSH variability in the SPNA north of 52°N, except the southern part of the Iceland Basin and the Rockall Trough, where the eddy variability associated

with the NAC branches is relatively strong.



## 4.2 Interannual SSH variability in the subpolar North Atlantic

The interannual variability of SSH averaged over 5°-45°W and 55°-65°N in the SPNA (green rectangle in Figure 1) was studied by Chafik et al. (2019). When excluding year-to-year variations, SSH increased by about 6 cm in 1993-2005, decreased by the same amount in 2006-2015, and then increased again by about 5 cm in 2016-2020 (black curve in Figure 6). Clearly, neither the $EOF_1$ (the SSH tripole; blue curve in Figure 6) nor the $EOF_2$ (red curve in Figure 6) alone can adequately explain the SSH variability in the SPNA. However, their sum (dotted curve in Figure 6) is sufficient to reasonably reconstruct SSH in the area averaged over 5°-45°W and 55°-65°N (green rectangle in Figure 1). The reconstructed SSH ($SSH_R$) computed with $EOF_1$ and $EOF_2$ matches the observed SSH well, with a correlation of 0.96 (compare solid and dotted curves in Figure 6), meaning that $SSH_R$ explains about 92% of the SSH variance. Individually, the $EOF_1$ and $EOF_2$ modes explain 44% and 48% of the SSH variance, respectively.

The part of the SPNA (5°-45°W and 55°-65°N) considered in Chafik et al. (2019) includes dynamically distinct basins. For example, the Iceland Basins and the Rockall Trough are the main pathways for the NAC in the SPNA, and these regions are characterized by elevated eddy kinetic energy. In contrast, eddy activity in the Irminger Basin is weaker by about a factor of three (e.g., Volkov, 2005). On the other hand, the Irminger Basin is one of the regions where deep water is formed due to the occurrence of wintertime deep convection (e.g., Pickart et al., 2003). Therefore, SSH averaged over all these basins can potentially mask mechanisms that drive the SSH variability, because different dynamical regimes are mixed when deriving such a mean.

In order to illustrate the relative contribution of temperature and salinity changes to the interannual-to-decadal variability of SSH in the SPNA, we compute the averages of $SSH_{ST}$, $SSH_T$, and $SSH_S$ over two regions with distinct dynamical processes (outlined by orange rectangles in Figure 1): (i) the eastern SPNA (0-30°W and 55-65°N), including the Iceland Basin, Rockall Plateau, and the Rockall Trough (Figure 7a); and (ii) the western SPNA (30-60°W and 53-65°N), including the Irminger Basin and the Labrador Sea (Figure 7b). The variability of SSH in both the eastern SPNA and the western SPNA is mostly steric in nature, as $SSH_{ST}$ in these regions explains 94% and 92% of the SSH variance, respectively (compare black and blue curves in Figure 7). In 2005-2015, $SSH_{ST}$ decreased by about 6 cm in the eastern SPNA and 5 cm in the western SPNA (blue curves in Figure 7). The decrease of $SSH_{ST}$ was caused by cooling and the associated reduction of $SSH_T$ by about 12 cm in the eastern SPNA and 6 cm in the western SPNA (red curves in Figure 7). The reduction of $SSH_T$ in both the eastern SPNA and western SPNA was compensated by freshening induced increase of $SSH_S$ by about 5 cm and 1 cm, respectively (green curves in Figure 7). It is interesting to note that while the sign of $SSH_{ST}$ anomalies is determined by $SSH_T$, the contribution of $SSH_S$ is substantial and comparable to the contribution of $SSH_T$, especially in the eastern SPNA.

## 4.3 Propagation of SSH anomalies

We have shown that SSH in the SPNA is adequately reconstructed by the two leading EOF modes. If the observed phase difference between these statistical modes is due to signal propagation, then both $EOF_1$ and $EOF_2$ describe the same physical





process at different stages of its evolution. Indeed, the application of the CEOF analysis presented below shows that $EOF_1$ and $EOF_2$ resemble the real and imaginary parts of $CEOF_1$, respectively. Recall that the real and imaginary parts of $CEOF_j$ can be

interpreted as the propagating signal at two stages separated by a quarter cycle. Displayed in Figure 8 is the time evolution of the spatial pattern of SSH reconstructed with $CEOF_1$ mode for one full cycle at phase stages separated by 45°. Note that SSH patterns at phases ±180° and 90° are almost identical to $EOF_1$ and $EOF_2$, respectively (Figure 2 a,b). Likewise, illustrating the temporal evolution of $CEOF_1$ mode, the real part of $CPC_1$ when rotated by 180° (Figure 9, dashed blue curve) corresponds to $PC_1$ (Figure 3, blue curve) and the imaginary part of $CPC_1$ (Figure 9, red curve) corresponds to $PC_2$ (Figure 3, red curve).

The reconstruction of the $CEOF_1$ mode (Figure 8) shows that the tripole-related sea-level variations in the tropical and subtropical bands resemble a standing wave. Negative and positive SSH in the tropical and subtropical bands at phase 0° are gradually replaced by positive and negative SSH at phase 180°, respectively, without a notable sign of signal propagation. On the other hand, a signal propagation is evident in the subpolar band of the tripole. In the first half of the cycle at phases -180° and -135°, negative anomalies start emerging near the eastern boundary, when positive anomalies are mainly concentrated in

the Irminger Basin and Labrador Sea. As the negative anomalies propagate towards the Irminger Basin and the Labrador Sea, the positive anomalies extend south-southwestward along the North America east coast and eventually leak into the subtropical gyre. SSH anomalies in the subtropical gyre then switch sign from negative to positive, which ultimately leads to the emergence of small positive SSH anomalies along the European coast seen at phase 0°. These anomalies are intensified at phase 45° and propagate westward peaking in the Iceland Basin at phase 135°, in the Irminger Basin at phase 135-180°, and in the Labrador

Sea at phase 180°, completing the full cycle.

As demonstrated by the real and imaginary $CPC_1$ time series (blue and red curves in Figure 9, respectively) and the temporal phase (dotted black curve in Figure 9), there were almost three full cycles of the decadal gyre-scale sea-level changes that exhibited signal propagation in the SPNA in 1993-2020. One full cycle is associated with the temporal phase change from −180° to 180°. In the northeastern SPNA, the first cycle started with a low SSH in 1993 (red curve) and it progressively reached

high SSH in 1996, corresponding to phases −90° and 90° in Figure 8, respectively. This signal propagated westward, with SSH reaching maximum values in the Irminger Basin and Labrador Sea in 1998 (dashed blue curve), corresponding to phase 180° in Figure 8. Counting from the first SSH minimum in the northeastern SPNA in 1993 (red curve) to the second SSH minimum in the western SPNA in 2000-2001 (dashed blue curve), the first cycle lasted 7-8 years. The second cycle started with a low SSH in the northeastern SPNA in 1999 (second minimum in the red curve) and ended with a local (third) SSH minimum in

2008-2009 in the western SPNA (dashed blue curve), thus taking 9-10 years to complete. Based on the temporal phase of the $CPC_1$ (dotted black line in Figure 9), there was no apparent propagation in 2006-2009. The third cycle can be counted from a local (third) SSH maximum in the northeastern SPNA in 2009 (red curve) that reached the Labrador Sea in 2011 (dashed blue curve). The second SSH maximum of this cycle has not yet been reached by the end of the record in 2020, so its overall duration has exceeded 10 years.



### 4.4 Relationship between sea level and wind forcing

The maximum correlation between the $PC_1$ (solid blue curve in Figure 3), showing the time evolution of the North Atlantic SSH tripole, and the low-pass filtered NAO index (shaded area in Figure 3) is -0.73, with the NAO leading by 10 months (the 95% significance level for correlation is about 0.45). Regression of SLP and 10-m winds on the $PC_1$ displays a familiar NAO dipole pattern with the subtropical high and subpolar low SLP centers (Figure 10a). Stronger/weaker westerly winds in the midlatitude North Atlantic associated with stronger/weaker subtropical high and subpolar low lead to higher/lower sea levels in the subtropical North Atlantic and lower/higher sea levels in the SPNA. Interestingly, while correlation between the $PC_2$ and the NAO is not significant, regression of SLP and 10-m winds on the $PC_2$ (Figure 10b) also exhibits a dipole pattern similar to the NAO, but with somewhat shifted pressure centers. This suggests that both the $EOF_1$ and the $EOF_2$ of the low-pass filtered SSH are possibly driven by the same atmospheric process, but are associated with different phases of its evolution.

To support this latter argument, we also regressed the wind stress curl and 10-m winds on the real parts of $CPC_1$ rotated every 45° between ±180° thus yielding the full-cycle evolution of wind forcing patterns associated with the leading CEOF mode (Figure 11). The obtained patterns illustrate that the distribution of wind stress curl, as the subtropical high and the subpolar low centers change their position and intensity, is clearly related to the SSH patterns associated with the $CEOF_1$ (compare Figures 8 and 11). The positive/negative wind stress curl anomalies drive the near-surface Ekman divergence/convergence and lead to the upper-ocean cooling/warming (blue/red shading in Figure 11). These anomalies are generally associated with lower/higher sea levels (Figure 8). For example, the westward propagation of negative/positive SSH from the northwest European shelf towards Labrador Sea (Figure 8) clearly follows similar shifts in the position of positive/negative wind stress curl anomalies (Figure 11). On the one hand, this comparison suggests that sea level and the upper-ocean heat content in the SPNA result from the oceanic adjustment to persistent, lasting longer than a year, wind forcing, which is also supported by the lagged correlation between the NAO and $PC_1$ (Figure 3). On the other hand, oceanic feedback to atmospheric forcing is also possible on these time scales.

It should be noted that wind forcing patterns at phases ±180° and 90° are similar to the SLP and wind patterns associated with $PC_1$ and $PC_2$, respectively (Figure 10). The analysis presented here indicates that the NAO, which is correlated with the $PC_1$ only, is not a sufficient proxy of wind forcing over the SPNA. The definition of the NAO is limited by an assumption that it is a standing oscillation pattern with the fixed subtropical high and the subpolar low-pressure centers. We have shown that the dynamic sea level variability in the SPNA is linked to the NAO-like atmospheric circulation patterns that change their location and intensity. The importance of accounting for the position and intensity of atmospheric center of action for determining causal relationships within the coupled ocean-atmosphere system in the North Atlantic has been reported earlier (e.g., Hameed and Piontkovski, 2004; Hameed et al., 2021).



## 4.5 Relationship between sea level and surface buoyancy forcing

To investigate the impact of surface buoyancy forcing on the interannual-to-decadal changes of SSH, we select four time intervals characteristic for the main tendencies reflected by $PC_1$ and $PC_2$ in Figure 3 (by the real and imaginary $CPC_1$ in Figure 9): 1994-2010 and 2011-2015 for $PC_1$ (real $CPC_1$), and 2004-2014 and 2015-2019 for $PC_2$ (imaginary $CPC_1$). For these time intervals, we compute the absolute changes of SSH, $SSH_{ST}$, $SSH_T$, and $SSH_S$ (Figures 12-15, a-d) and the contributions to these changes driven by surface heat (Figures 12-15, e-i) and freshwater fluxes (P+E; not shown). Because the SSH changes driven by surface freshwater fluxes appeared to be three orders of magnitude smaller than those driven by surface heat fluxes, they are not considered in the following. The impact of surface freshwater fluxes on the regional $SSH_S$ changes (Figure 12-15, d) is also negligible, thus, suggesting that these changes are mainly driven by the advection of freshwater.

The overall increase of sea level in the Labrador Sea and Irminger Basin in 1994-2010 associated with the North Atlantic SSH tripole is well explained by the $SSH_{ST}$ change (Figure 12 a,b). Only over the northern part of the Iceland Basin and over the Rockall Plateau and Rockall Trough, the $SSH_{ST}$ change exceeds the total SSH change. The differences between the changes of SSH and $SSH_{ST}$ in 1994-2010 can be due to the lack of in-situ temperature measurements prior to the start of the widespread use of Argo floats in the region. The spatial pattern and the sign of the observed $SSH_{ST}$ change are determined by the $SSH_T$ change (Figure 12c), which is partly balanced by the $SSH_S$ change (Figure 12d). The net surface heat flux anomalies ($Q_{NET}$) can explain the SSH/$SSH_{ST}$ increase in the Labrador Sea and over the Rockall Plateau and the SSH/$SSH_{ST}$ decrease in the southern part of the Iceland Basin (Figure 12e). The shortwave ($Q_{SR}$) and thermal ($Q_{TR}$) radiation anomalies largely compensated each other (Figure 12f,g), and the largest contribution to $Q_{NET}$ anomalies in 1994-2010 came from the sensible ($Q_{SH}$) and latent ($Q_{LH}$) heat flux anomalies. The observed warming in the Labrador Sea in 1994-2010 was mainly caused by the $Q_{SH}$ anomaly. This implies that there were tendencies for the ocean to be colder than the air aloft and for the air temperature just above the surface to be increasing upward. This pattern is typical during periods with an anomalous heat flux into the ocean (positive $Q_{SH}$ anomaly). The secondary contribution to the Labrador Sea warming came from the positive $Q_{LH}$ anomaly, also suggesting that the air temperature was higher than the ocean temperature and the humidity was sufficient to cause condensation of water vapor on the surface of the ocean. This resulted in a loss of heat from air into the ocean (positive $Q_{LH}$ anomaly). In the northern part of the Irminger Basin, the increase of SSH/$SSH_{ST}$ in 1994-2010 (Figure 12a,b) was associated with the concurrent heat loss to the atmosphere (Figure 12e), which must have been compensated by an increased oceanic heat advection into the region.

The 2004-2014 period was characterized by a decade-long decrease of SSH over most parts of the northeastern SPNA (Figures 2b, solid red curves in Figures 3 and 9), in particular in the Iceland and Irminger Basins (Figure 13a). This decrease observed by satellite altimetry corresponds well to the decrease of $SSH_{ST}$ (Figure 13b), which was mainly determined by the ocean cooling reflected in the $SSH_T$ change (Figure 13c). The latter was partly compensated by freshening-induced $SSH_S$ increase in the eastern part of the SPNA (Figure 13d). The $Q_{NET}$ anomaly (Figure 13e) can explain the 2004-2014 cooling and SSH decrease only in the northeastern SPNA. In the interior of the Labrador Sea, while the SSH was decreasing, the atmosphere




was warming the ocean (Figure 13e), which implies an increased oceanic heat transport out of the basin. The contributions of $Q_{SR}$ and $Q_{TR}$ anomalies to the $Q_{NET}$ anomaly are rather small and compensate for each other (Figure 13f,g). It is interesting to

note that unlike during the 1994-2010 period (Figure 12f,g) the contributions of $Q_{SH}$ and $Q_{LH}$ anomalies are geographically distinct (Figure 13f,g). The $Q_{SH}$ anomaly makes the largest contribution to the positive $Q_{NET}$ anomaly in the western SPNA (Figure 13h), meaning that in this region the ocean temperature relative to the air temperature was colder than usual. This led to an anomalous heat flux from the atmosphere into the ocean. In contrast, in the northeastern SPNA, the $Q_{LH}$ anomaly was the largest contributor to the negative $Q_{NET}$ anomaly (Figure 13i). In the latter case, the ocean temperature relative to the air

was warmer than usual, thus favoring evaporation and the associated upper-ocean cooling.

In 2011-2015, a tripole-related decrease of SSH occurred over most parts of the SPNA, with the largest anomalies in the Irminger Basin and the Labrador Sea (Figure 14a). This decrease was mainly steric in nature (Figure 14b), and it was associated with strong cooling (Figure 14c) and freshening (Figure 14d). It is interesting to note that in most regions the $Q_{NET}$ anomaly did not provide the largest contribution to the observed SSH change in 2011-2015 (Figure 14d). Only its contributions over

the northwest European shelf, in the Norwegian Sea, and along the East Greenland and Labrador Currents were substantial. This means that the oceanic advection of heat and freshwater was the major driver for the 2011-2015 decrease of SSH in most parts of the SPNA. The distributions of $\Delta SSH_T$ (Figure 14c) and $\Delta SSH_S$ (Figure 14d) suggest that advection was mainly associated with the NAC. The structure of the $Q_{NET}$ anomalies was mostly determined by the nearly equal contributions from $Q_{SH}$ and $Q_{LH}$ anomalies with the strongest impacts along the East Greenland and Labrador Currents (Figure 14h,i). The $Q_{SR}$

and $Q_{TR}$ anomalies were somewhat smaller and largely compensating each other (Figure 14f,g). We recall that an exceptionally cold anomaly, termed the 'cold blob', occurred in the SPNA in 2015 (Ruiz-Barradas et al., 2018; Chafik et al., 2019). The analysis presented here shows that while the onset of this strong cooling started in 2004 in the ESPNA (red curve in Figure 7a) and was partly driven by the negative $Q_{NET}$ anomalies in 2004-2014 (Figure 13e), the advection of colder and fresher water was apparently the main driver for the cold and fresh anomaly in 2011-2015 (Figure 14 c,d).

During the following 2015-2019 period, SSH was rising in the northern and northeastern parts of the SPNA, particularly along the bottom topographic features associated with the Rockall Plateau and Reykjanes Ridge and along the Greenland shelf (Figure 15a). Mostly, the observed SSH increase adequately compares to the $SSH_{ST}$ increase (Figure 15b). What is interesting to note is that this increase was not mainly determined by the $SSH_T$ change as in the previous time intervals. The upper-ocean warming was leading to the SSH rise only in the Iceland Basin, over the Rockall Plateau, and further south upstream of the

NAC (Figure 15c). Warming in these areas was in part driven by the positive $Q_{NET}$ anomalies (Figure 15e) and by the advection of warmer and saltier water by the NAC (Figure 15 c,d). The positive $Q_{NET}$ anomalies were also observed over the Reykjanes Ridge and in the Irminger Basin. However, these anomalies did not lead to the upper-ocean warming and they were accompanied by a negative $SSH_T$ tendency (compare Figures 15e and 15c), which is suggestive of the oceanic heat flux out of these areas. An anomalous heat loss to the atmosphere along the East Greenland Current (Figure 15e), which was, however,

associated with the local SSH increase, also suggests the dominant role of oceanic heat advection within the area influenced



by the current. The largest contribution to the observed SSH rise in the interior of the Irminger Basin and Labrador Sea in 2015-2019 was provided by the upper-ocean freshening and the associated $SSH_S$ rise (Figure 15d). Because the impact of surface freshwater flux (P+E; not shown) is negligible, the observed freshening could only be driven by the advection of freshwater and/or the continental runoff, including the meltwater from Greenland glaciers. Similar to the previous time
intervals, the contribution of the $Q_{SR}$ and $Q_{TR}$ anomalies to the $Q_{NET}$ anomaly was small and balancing each other (Figure 15f,g). The $Q_{NET}$ anomaly was mainly driven by the $Q_{SH}$ and $Q_{LH}$ anomalies, with the largest contribution from the latter (Figure 15h,i). Apparently, there was a tendency for the ocean to be colder than the atmosphere, thus, favoring anomalous heat flux from the air into the ocean.

### 4.6 The role of the large-scale ocean circulation in the SPNA

It is well known that persistent wind forcing leads to changes in the upper ocean heat content through Ekman dynamics and the adjustment of the large-scale geostrophic circulation. In Section 4.4, we showed that the observed westward propagation of sea level anomalies in the SPNA is consistent with shifts in wind stress curl patterns. In the previous section, we demonstrated that while some of the observed regional tendencies of sea level can be explained by surface heat fluxes, there are regional tendencies that can only be accounted for by the advection of heat and freshwater. For example, the strong decrease
of SSH/$SSH_{ST}$ in 2011-2015 ('cold blob') was mainly caused by the advection of colder and fresher water masses (Figure 14). In this section, we present the large-scale ocean circulation in the SPNA deduced from Argo trajectories at 1000-bar depth (Figure 16a) and from eddy propagation velocities calculated from satellite altimetry data (Figure 16b), and we qualitatively analyze how ocean circulation could contribute to the observed westward propagation of sea level anomalies during the observational period.

The time-mean circulation at 1000-dbar depth shows a distinct cyclonic pattern constrained by bottom topography (Figure 16a). This pattern is similar to the schematic upper-ocean circulation shown in Figure 1 based on previous studies (e.g., Schmitz and McCartney, 1993; Schott and Brandt, 2007). The velocities derived from Argo trajectories well correspond to those derived from hydrographic sections and satellite altimetry (e.g., Sarafanov et al., 2012). The flow associated with the NAC, once it reaches the northern part of the Iceland Basin, recirculates southwestward along the eastern flank of the Reykjanes Ridge with
the speeds of about 5 cm s$^{-1}$. Upon crossing the Reykjanes Ridge, it follows its western flank, finally joining the system of western boundary currents composed of the East and West Greenland Currents and the Labrador Current, where the speeds often exceed 15 cm s$^{-1}$. The eddy propagation pattern and speeds (Figure 16b) are similar to the 1000-dbar velocities, meaning that eddies follow the mean SPNA circulation pathways, steered by bottom topography. Because the eddy propagation velocities (Figure 16b) are directly derived from the SSH anomalies, they are representative for property transports in the upper
ocean. Therefore, heat and freshwater signals advected to or generated in the Iceland Basin are transferred first to the Irminger Basin and then to the Labrador Sea. Assuming the average eddy propagation speed of 5 cm s$^{-1}$ (Figure 16b), it takes about 1.5 years for a SSH signal to propagate from the northern part of the Iceland Basin (~18°W, 62°N) to Cape Farewell and about 2



years to reach the northern part of the Labrador Sea at 63°N following the mean circulation pathway along the eastern and western flanks of the Reykjanes Ridge and the Greenland continental shelf. This is the same time scale as the one we identified earlier using the CEOF analysis (see Section 4.3). Therefore, it is reasonable to suggest that the advection of heat and freshwater by the mean ocean circulation in the SPNA is a possible mechanism for the observed east-to-west propagation of SSH anomalies.

To further illustrate the role of advection, we present the time-depth diagrams of potential temperature and salinity anomalies from EN4 at three selected locations in the Iceland Basin (60°N, 20°W), in the Irminger Basin (60°N, 35°W), and in the Labrador Sea (58°N, 50°W) (Figure 17). It should be noted that the EN4 data is more reliable after the Argo array achieved its complete coverage in the Atlantic in 2003. The anomalies that dominate the observed interannual-to-decadal SSH variability extend down to 1500-2000 m. Potential temperature anomalies (Figure 17 a-c) are driven by both the surface heat fluxes and the advection of heat. Because the impact of surface freshwater fluxes is very small (see Section 4.5), salinity anomalies (Figure 17 d-f) are mainly due to the advection of freshwater and continental runoff. The time-depth diagrams indicate that some anomalies are first observed in the Iceland Basin and then they reach the Irminger Basin and the Labrador Sea. This is particularly evident in salinity anomalies, which are more representative for the impact of advection than temperature anomalies. For example, the negative potential temperature anomaly observed in the Iceland and Irminger Basins in 1994-1995 reaches the Labrador Sea in 1-2 years. The 2015-2018 strong cooling anomaly in the Iceland Basin peaks in the Irminger Basin in 2017. The near-surface positive salinity anomaly observed in the Iceland Basin in 1998 reaches the Irminger Basin in 1999 and the Labrador Sea in 2000. The strong upper-ocean freshening observed in the Iceland Basin in 2015-2020 also reached the Irminger Basin 1-2 years later, and it only started to show up as subsurface freshening in the Labrador Sea in 2020.

**5 Conclusions**

This study presents a reconsideration of the interannual-to-decadal SSH variability in the North Atlantic, the leading mode of which exhibits a tripole pattern (Volkov et al., 2019b). This tripole was originally detected by the conventional EOF analysis (Figure 2 a,c; blue curve in Figure 3), which is effective at depicting only standing oscillations. The sign of the tripole is mainly determined by $SSH_T$ (Figure 4a), partly balanced by a sizable contribution from $SSH_S$ (Figure 4d). The analysis presented here demonstrates that the first EOF mode alone does not adequately represent the interannual-to-decadal sea level variability in the SPNA. We show that the first mode explains the majority (60-90%) of the interannual SSH variance only in the Irminger Basin and in the Labrador Sea (Figure 5a), while the second EOF mode accounts for 60-80% of the interannual SSH variance in the northeastern SPNA (Figure 5b). This means that only the combination of the two leading modes represents the SSH variability in the SPNA adequately (Figure 5c).

Furthermore, we demonstrate that the two modes do not represent two distinct physical processes. Instead, they belong to the same process and arise due to the general east-to-west propagation of SSH anomalies (Figure 8). The CEOF analysis, which



is designed to detect propagating (as opposed to standing) signals, yields the real and imaginary parts of the leading CEOF
mode that correspond well to the first and the second EOF modes, respectively. This suggests that the two leading EOF modes
evolve as a quadrature pair associated with a propagation of SSH anomalies. Based on the CEOF analysis, there were almost
three full cycles of the tripole-related SSH changes that exhibited westward propagation of SSH anomalies in the SPNA in
1993-2020 (Figure 9). The reconstruction of the leading CEOF mode at different phases of one full cycle shows that SSH
anomalies first appear along the northwestern European shelf and then gradually propagate westward (Figure 8). It takes about
2 years for a signal to travel from the Iceland Basin to the Labrador Sea, and it takes 7-10 years for one full cycle to complete.

It has been documented that the North Atlantic SSH tripole is correlated with the NAO: stronger/weaker than average mid-
latitude westerly winds associated with positive/negative NAO phases lead to divergence/convergence and lower/higher sea
levels in the SPNA (Volkov et al., 2019).  To expand on the earlier work, we analyzed what mechanisms were responsible for
the observed interannual-to-decadal SSH variability in the SPNA in 1993-2020, and what mechanisms could be responsible
for the observed signal propagation. We find that since the two leading EOF modes depict the same physical process, the
evolution of the second EOF mode is also related to an NAO-like dipole SLP pattern, but with shifted atmospheric pressure
centers (Figure 10). The definition of the NAO implies that it is a stationary standing oscillation pattern. However, the
subtropical high- and the subpolar low-pressure centers in the North Atlantic change both their intensity and position (e.g.,
Hameed and Piontkovski, 2004; Hameed et al., 2021). Therefore, we conclude that both the first and the second EOF modes
reflect oceanic response to atmospheric forcing at different phases of its evolution. This conclusion is supported by the space
and time evolution of wind forcing patterns associated with the first CEOF mode of the low-frequency SSH, which shows that
wind stress curl anomalies are clearly associated with SSH anomalies of the respective sign (Figure 11). This leads us to
another conclusion that persistent (longer than a year) wind forcing patterns, at least partly, drive the observed interannual-to-
decadal SSH variability in the SPNA. As the wind forcing patterns change their positions, they generate associated shifts of
SSH anomalies.

The role of surface buoyancy forcing over the SPNA in driving the interannual-to-decadal changes of SSH, $SSH_T$, and $SSH_S$
is investigated over several characteristic time intervals: 1994-2010, 2004-2014, 2011-2015, and 2015-2019 (Figures 12-15).
The impact of surface freshwater fluxes is found to be negligible in all periods, so that any changes in $SSH_S$ are mainly driven
by the advection of freshwater. Advection was apparently the main driver for the strong upper-ocean freshening observed in
the eastern SPNA after 2010 (Figure 7a, Figure 13d). We find that the net surface heat flux anomalies, mainly caused by the
sensible and latent heat flux anomalies, can fully or partly explain some regional tendencies in $SSH_T$. For example, $Q_{NET}$
anomalies drove the increase of $SSH_T$ in the Labrador Sea in 1994-2010 and in the Iceland Basin in 2015-2019 (Figure 12 c,e),
and the decrease of $SSH_T$ in the northeastern SPNA in 2004-2014 (Figure 13 c,e) and in the Labrador Sea in 2011-2015 (Figure
14 c,e). However, there are regions and time periods when changes in $SSH_T$ can only be explained by advection, the
contribution of which cannot not be directly estimated from observations. These are the regions and periods, in which changes
in $SSH_T$ are either much larger or have a sign opposite to the changes implied by the $Q_{NET}$ anomalies. A prominent example is



a strong cooling in the SPNA in 2011-2015, known as the 'cold blob', which coincided with contemporary freshening (Figure 13 c,d). This period was apparently characterized by the advection of colder and fresher water masses into the region, consistent with findings of Holliday et al. (2020).

In addition to shifting wind stress curl patterns, the observed westward propagation of SSH anomalies could be caused by the mean ocean circulation in the SPNA. The likely role of ocean currents is qualitatively assessed using the climatological velocities at 1000-dbar obtained from Argo trajectories (Figure 16a) and eddy propagation velocities estimated from satellite altimetry measurements (Figure 16b). Both velocities depict the cyclonic circulation in the SPNA, constrained by bottom topography. The eddy propagation velocities signify the propagation of SSH anomalies and, therefore, they are characteristic
for the depth-integrated temperature and salinity (that define density and, consequently SSH) transports. We find that the time required for SSH anomalies to propagate from the Iceland Basin to the Labrador Sea (1-2 years) is consistent with the time implied by the velocity estimates and by the upper 2000-m temperature and salinity anomalies (Figure 17). This means the observed westward propagation of SSH anomalies is partly due to the mean direction of ocean currents in the SPNA. Any anomaly generated locally by atmospheric forcing or advected from another region is ultimately carried towards the Labrador
Sea by ocean currents.

We conclude that the observed interannual-to-decadal variability of SSH, including the westward propagation of SSH anomalies, is the result of a complex interplay between the local wind and surface buoyancy forcing, and the advection of properties by mean ocean currents. The relative contribution of each forcing term to the variability is space and time dependent. Our findings suggest that the concept of the North Atlantic SSH tripole as the leading EOF mode of the interannual SSH
variability introduced in Volkov et al. (2019) needs to be reconsidered, at least with respect to its application in the SPNA. While the standing oscillation of the interannual-to-decadal SSH anomalies is a reasonable approximation of the variability in the subtropical and tropical North Atlantic, signal propagation needs to be accounted for in the SPNA. Therefore, the tripole in the SPNA needs to be either based on the leading CEOF mode or on the first two EOF modes combined. The observed east-to-west propagation of SSH anomalies in the SPNA suggests the potential predictability of SSH changes and conditions
favoring deep convection events in the region. This study puts the interannual-to-decadal changes of SSH in the SPNA in a broader context of the gyre-scale SSH variability in the entire North Atlantic.

**Acknowledgements**

The delayed-time satellite altimetry maps are processed and distributed by the Copernicus Marine and Environment Monitoring Service (http://marine.copernicus.eu; https://doi.org/10.48670/moi-00148). EN.4.2.2 data were obtained from
https://www.metoffice.gov.uk/hadobs/en4/ and are © British Crown Copyright, Met Office, provided under a Non-Commercial Government License http://www.nationalarchives.gov.uk/doc/non-commercial-government-licence/version/2/. Argo data were collected and made freely available by the International Argo Program and the national programs that



contribute to it. (http://www.argo.ucsd.edu, http://argo.jcommops.org). The Argo Program is part of the Global Ocean
Observing System. Hersbach, H. et al., (2019) was downloaded from the Copernicus Climate Change Service (C3S) Climate
Data Store (doi:10.24381/cds.f17050d7). This research was supported by NOAA's Climate Variability and Predictability
program (grant number NA20OAR4310407) and by the NOAA Atlantic Oceanographic and Meteorological Laboratory, and
it was carried out in part under the auspices of the Cooperative Institute for Marine and Atmospheric Studies, a Cooperative
Institute of the University of Miami and NOAA, cooperative agreement #NA20OAR4320472. The authors are grateful to Dr.
Sang-Ki Lee for providing feedback on the initial version of the manuscript.

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



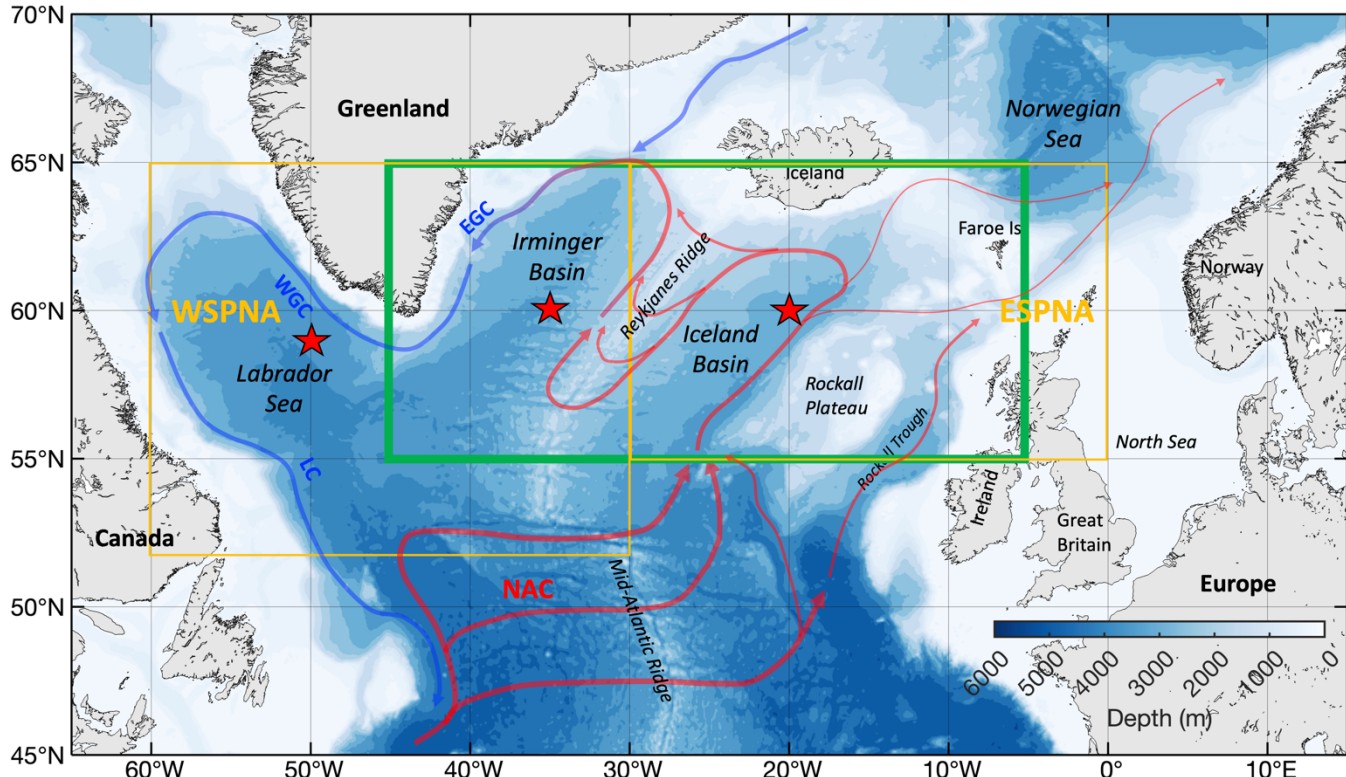

**Figure 1: Bottom topography and schematic upper-ocean circulation in the Subpolar North Atlantic. Abbreviations: NAC - North Atlantic Current, EGC - East Greenland Current, WGC - West Greenland Current, LC - Labrador Current. The green rectangle bounds the area used for averaging in Chafik et al. (2019). The orange rectangles bound the areas used for averaging in the Western SPNA (WSPNA) and in the Eastern SPNA (ESPNA). The red stars show the locations of temperature and salinity profiles shown in Figure 17.**





**Figure 2: EOF analysis of sea level anomalies: (a) EOF₁ and (b) EOF₂ modes of sea level anomalies measured by satellite altimetry;**
**(c) EOF₁ and (d) EOF₂ modes of steric sea level anomalies derived from EN4 temperature and salinity profiles.**





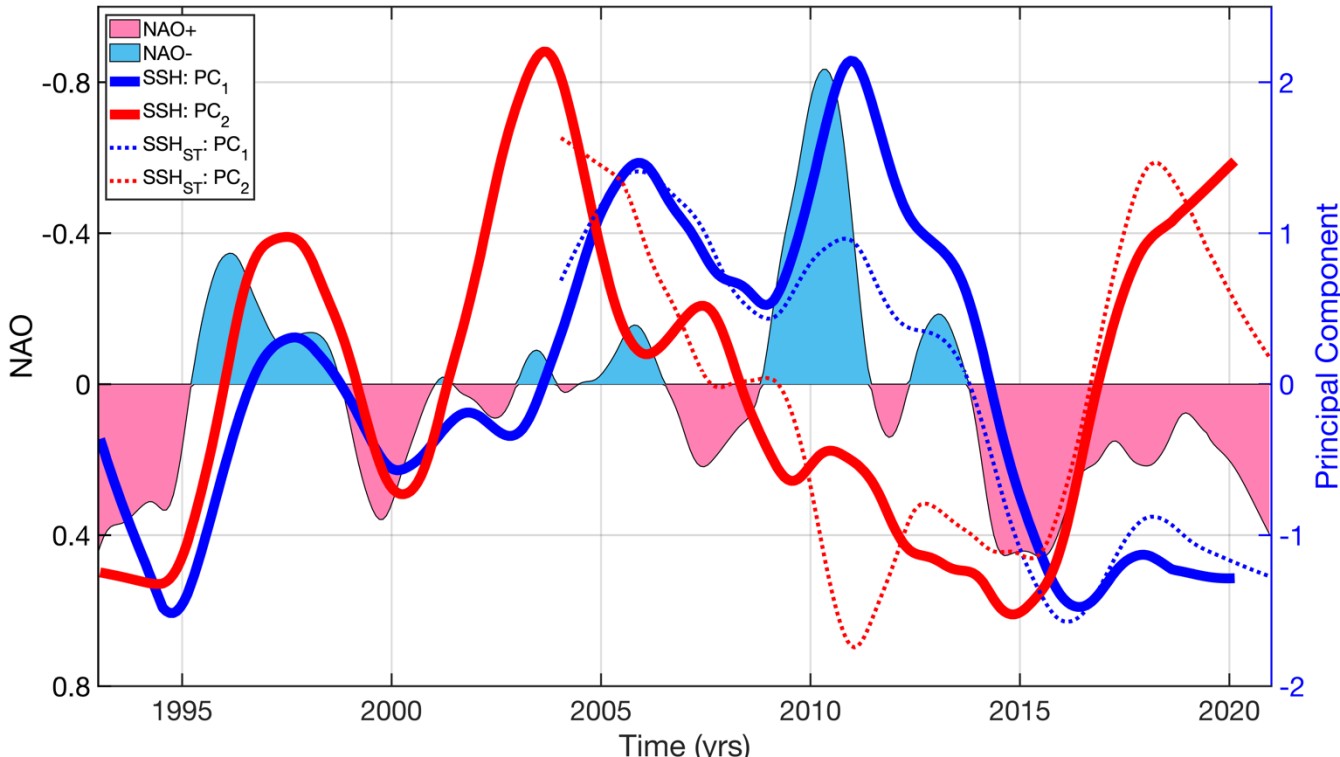

**Figure 3: Time evolution (principal components) of EOF$_1$ (blue curves) and EOF$_2$ (red curves) modes of SSH (solid curves) and SSL (dotted curves). The low-pass filtered monthly NAO index is shown by color shading: (pink) positive NAO and (blue) - negative NAO (note that the left axis for NAO is reversed).**



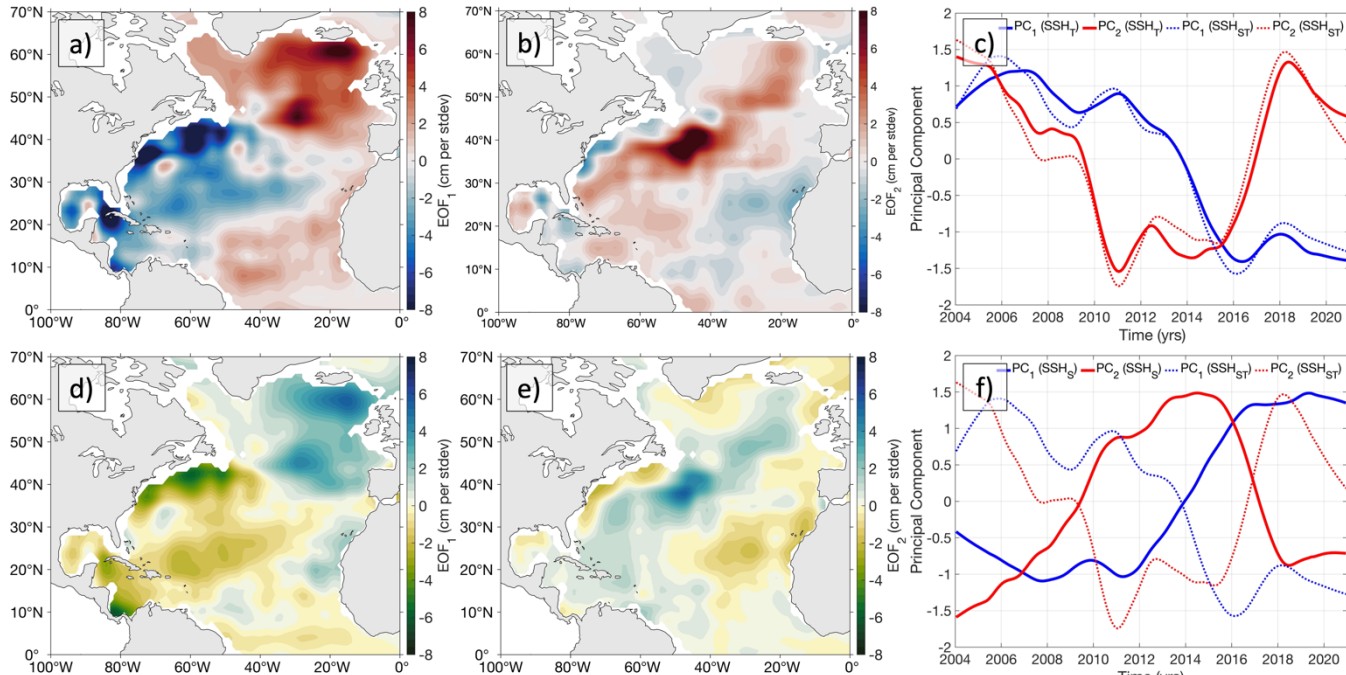

**Figure 4: EOF analysis of thermosteric (SSH$_T$) and halosteric (SSH$_S$) sea level derived from EN4 temperature and salinity profiles:**

**(a) EOF$_1$ and (b) EOF$_2$ modes of SSH$_T$, (c) PC$_1$ (blue) and PC$_2$ (red) of SSH$_{ST}$ (dotted) and SSH$_T$ (solid), (d) EOF$_1$ and (e) EOF$_2$ modes of SSH$_S$, (f) PC$_1$ (blue) and PC$_2$ (red) of SSH$_{ST}$ (dotted) and SSH$_S$ (solid).**






Figure 5: The portion of local variance of the low-pass filtered SSH explained by (a) EOF$_1$, (b) EOF$_2$, and (c) the composition of EOF$_1$ and EOF$_2$ in the subpolar gyre of the North Atlantic. The bathymetry contours are shown every 1000 m. The rectangles bound the areas of the Western SPNA (WSPNA) and Eastern SPNA (ESPNA) used for averaging the time series shown in Figure 7.





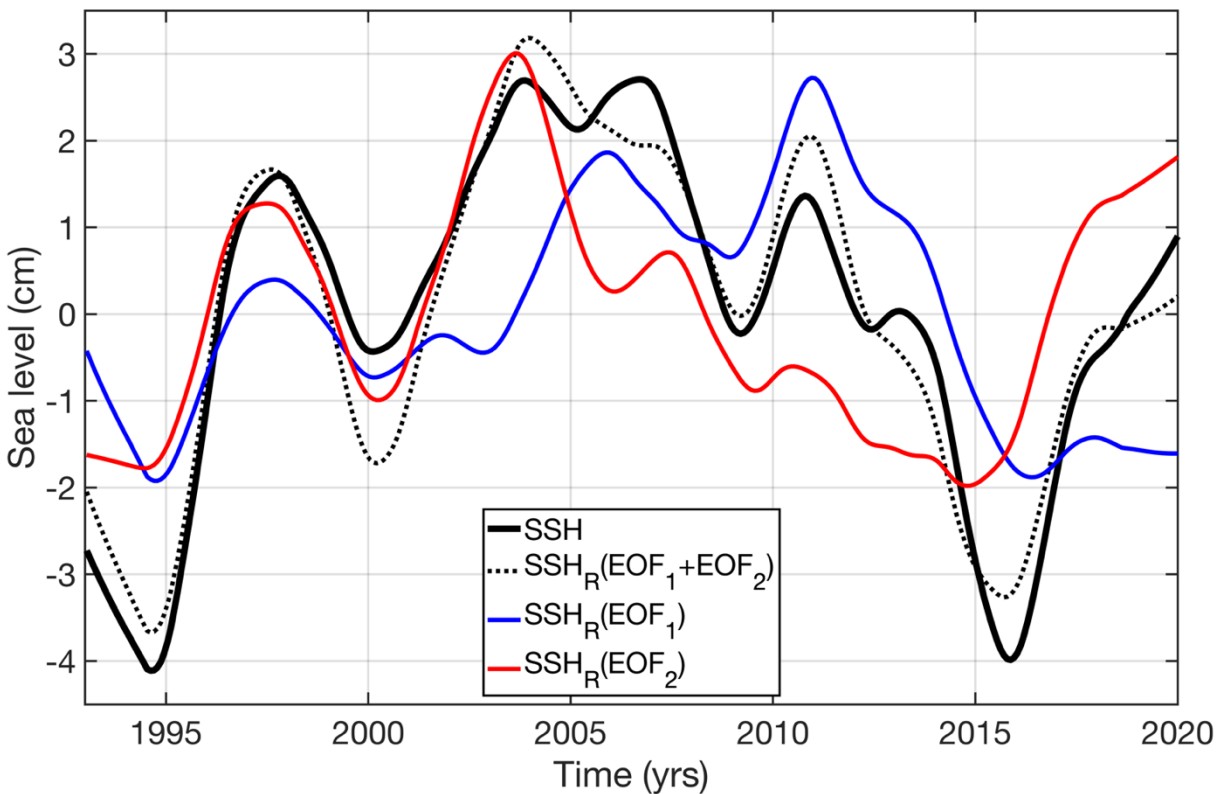

**Figure 6: Time series averaged over 5°-45°W and 55°-65°N (as in Chafik et al., 2019): detrended SSH (solid black), SSH$_R$ reconstructed using the combination of EOF$_1$ and EOF$_2$ (dotted black), SSH$_R$ reconstructed using EOF$_1$ (blue), and SSH$_R$ reconstructed using EOF$_2$ (red).**

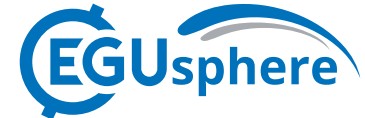

**Figure 7: Time series of SSH (black), SSH$_{ST}$ (blue), SSH$_T$ (red), and SSH$_S$ (green) averaged over (a) 0-30°W and 55-65°N in the Eastern SPNA (ESPNA) and (b) 30-60°W and 53-65°N in the Western SPNA (WSPNA) (the areas of ESPNA and WSPNA are shown in Figures 1 and 5).**





**Figure 8: Reconstruction of the CEOF₁ mode of the low-pass filtered SSH showing one full cycle (-180° – 180°) at 45° phase intervals. The angle of rotation is shown in the right lower corner of each panel. Note that CEOF₁ rotated at 180° and 90° is similar to EOF₁ and EOF₂, respectively.**






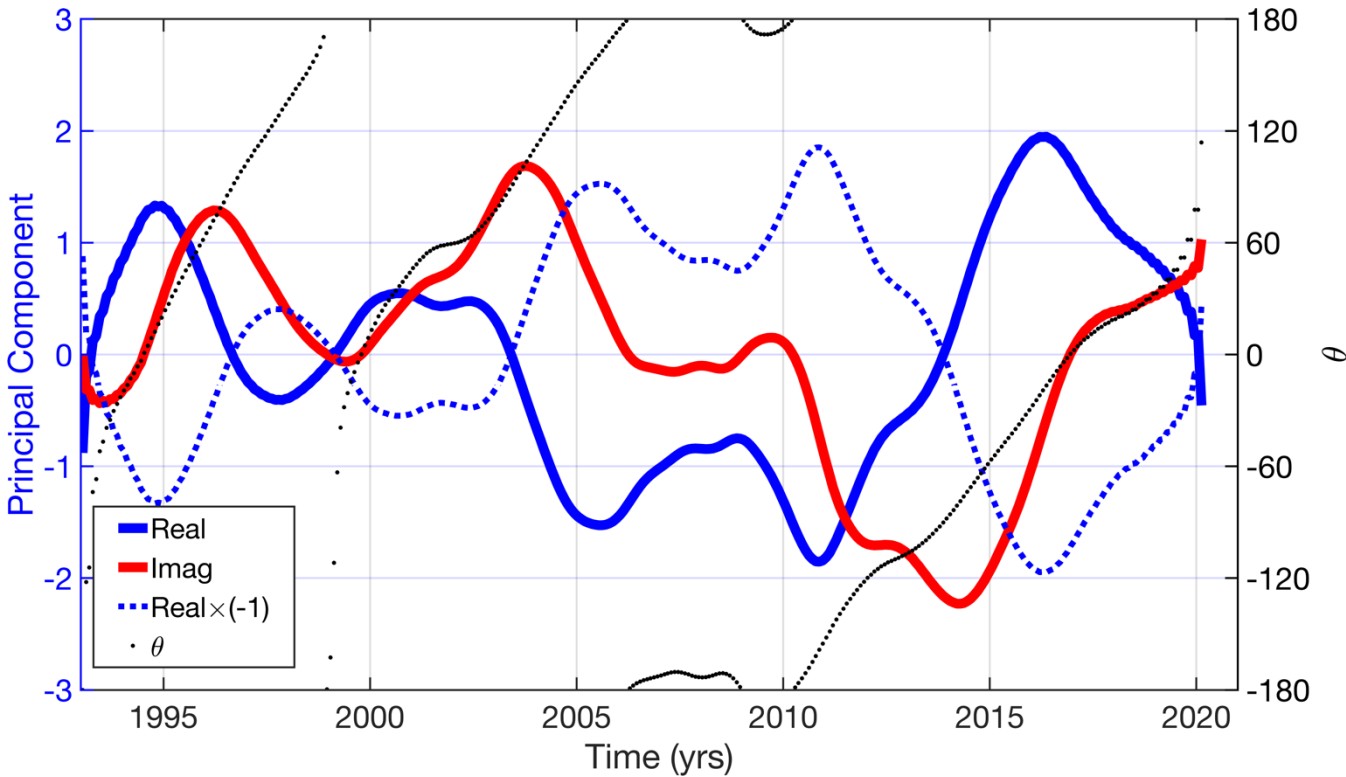

**Figure 9: Time evolution of CEOF$_1$ mode: (solid blue curve) real and (red curve) imaginary components, (dashed blue curve) real part of CPC$_1$ rotated by 180°, and (dotted black curve) the temporal phase of the mode.**






**Figure 10: Regression maps of (color) SLP and (arrows) 10-m wind velocity on (a) PC$_1$ and (b) PC$_2$; the units are Pa per standard deviation of PC$_j$ and m/s per standard deviation of PC$_j$, respectively.**





**Figure 11: Regression maps of (arrows) 10-m wind velocity and (color) wind stress curl on $CPC_1$ rotated every 45° between ±180° showing the full-cycle evolution of wind forcing patterns associated with the $CEOF_1$ mode of the low-pass filtered SSH. Positive/negative wind stress curl anomaly is associated with the upper-ocean cooling/warming (blue/red colors).**



**Figure 12: SSH change in 1994-2010: (a) total as observed with satellite altimetry, (b) steric, (c) thermosteric, (d) halosteric, (e) due to the net surface heat flux, (f) due to the shortwave radiation, (g) due to the thermal (longwave) radiation, (h) due to the sensible heat flux, and (i) due to the latent heat flux.**






**Figure 13: Same as Figure 11 but for the 2004-2014 period.**






**Figure 14: Same as Figure 11 but for the 2011-2015 period.**



**Figure 15: Same as Figure 11 but for the 2015-2019 period.**

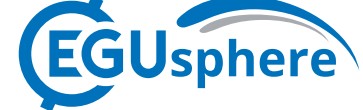

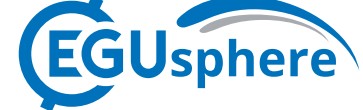

**Figure 16: (a) Velocities at 1000-dbar pressure level based on Argo float trajectories and Argo profiles of temperature and salinity; (b) eddy propagation velocities calculated from satellite altimetry data.**



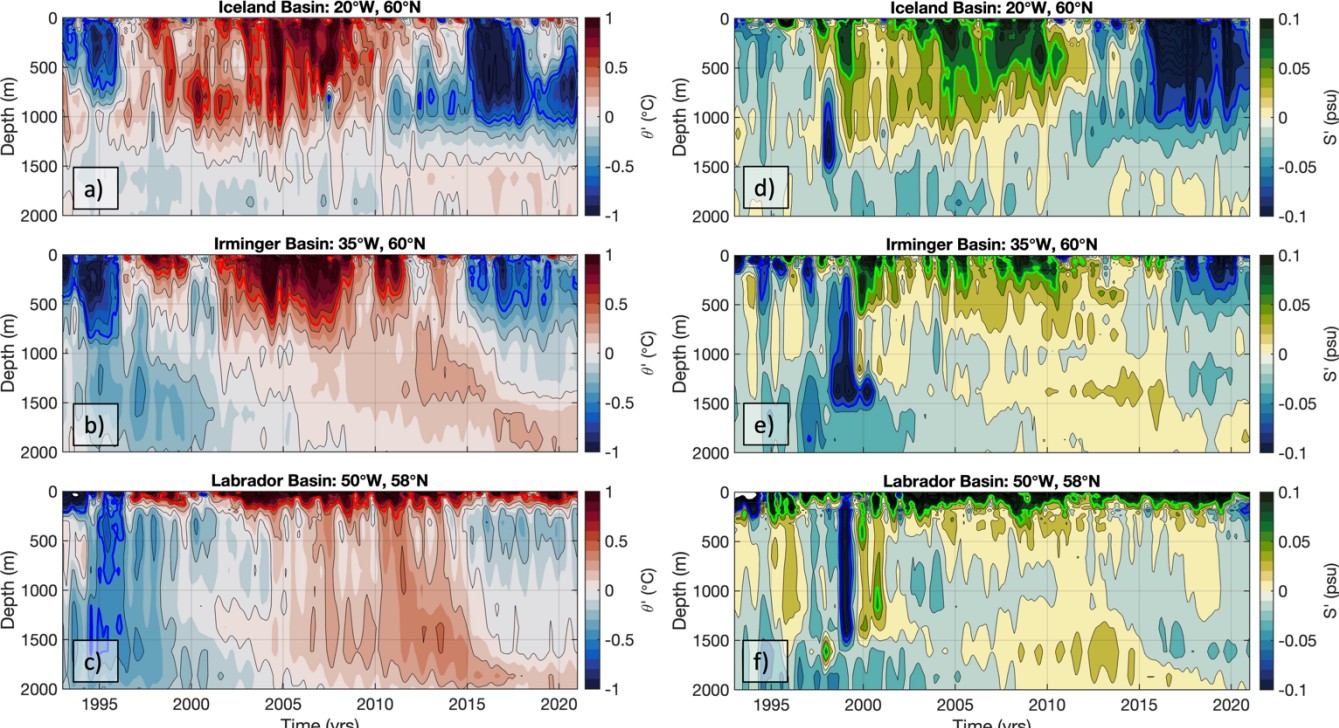

**Figure 17: Anomalies of (a-c) potential temperature and (d-f) salinity in (a,d) the Iceland Basin (at 20°W, 60°N), (b,e) the Irminger Basin (at 35°W, 60°N), and (c,f) the Labrador Basin (at 50°W, 58°N) (the locations are shown by red stars in Figure 1). The red and blue contours in a-c show 0.5 and −0.5°C isotherms, respectively. The green and blue contours in d-f show 0.06 and −0.06 psu.**