# Peer review of "Interannual to decadal sea level variability in the subpolar North Atlantic: The role of propagating signals"

_EGUsphere, 2022_

## Author Response (AR1)

**RC1:**

**General comments**

The study uses satellite and hydrographic data to characterize the inter-annual variability of SSH, focusing on the subpolar North Atlantic (SPNA). Empirical orthogonal function (EOF), and complex EOF (CEOF) analysis, of the North Atlantic SSH show that the first two EOF modes do not represent independent processes, but a signal propagating from east to west in the SPNA. The study then adds analysis of wind forcing and surface buoyancy fluxes to examine whether SSH variability and the east-west propagation is driven by winds, local air-sea fluxes or ocean advection, and relate the results to those of previous studies in the region. I enjoyed reading this and find the study to be well-written and convincing and to comprise useful insight into subpolar North Atlantic variability. Below I have a few specific queries, and a few minor technical issues.

We thank Dr. Fox for his time to review our paper, constructive comments, and for attention to details, in particular for finding errors in the equations. We have tried to address all concerns raised by Dr. Fox, and our answers to his comments are shown below in blue font.

**Specific comments**

Introduction and conclusions

Subpolar Gyre Index (SPGI)?

There exists extensive literature on the SPGI, which is often defined similarly, but not identically, to the EOFs used here. Generally, the SPGI had settled on the use of the first EOF (EOF1) of SSH variability (including the long-term trend I think) over the subtropical and subpolar gyres (about 30 to 65 N, contrasting to the larger area north of the equator up to 70N used here). Hatun and Chafik (2018,

https://onlinelibrary.wiley.com/doi/abs/10.1029/2018JC014101), show that more recently EOF1 in this region has become dominated by the largescale linear trend in the SSH, with subpolar gyre variability now in EOF2 (or a combination of EOF1 and EOF2). Superficially the first two EOFs in Hatun and Chafik appear similar to those presented here, but the time series in the principal components appear different. I think, given the focus on the SPNA, the present study needs more reference to the existing SPGI literature, highlighting the differences and similarities between 'classic' SPGI and the metrics used here, in both the introduction and in discussion. Does the study, for example, show that a single EOF is insufficient or inappropriate to characterise the major, propagating, variability in the SPNA?

This is an important remark, in response to which we have included relevant information to the Discussion and Conclusions (3rd paragraph):

"... It is necessary to mention that in order to describe the variability of SSH and ocean circulation in the SPNA, several authors have also used the 'subpolar gyre index', which, like the tripole, is based on an EOF decomposition of SSH fields (Häkkinen & Rhines, 2004; Hátún et al., 2005; Berx & Payne, 2017; Foukal & Lozier, 2017). The main difference between the subpolar gyre index and the tripole is that the global mean sea level is not subtracted from SSH

fields prior to the computation of the former. Therefore, the subpolar gyre index exhibits a trend characteristic of the global mean sea level rise. The regional dynamic changes are then represented by higher modes, which led Hátún & Chafik (2018) to justly conclude that  $PC_2$  in their calculation is a better metric for a gyre index than  $PC_1$ . Our results imply that with the signal propagation, if the global mean sea level is not subtracted prior to the EOF analysis, even the third mode needs to be considered."

**Section 4. Results**

I was occasionally confused about whether the authors were talking about their work in the current paper or previous published work by other authors. Examples are paragraph 1 of both sections 4.1 and 4.2. I think this may be due to a tendency to switch between describing results and figures in the past and present tenses.

**Thank you for noting this issue. In the revised manuscript, we have modified those paragraphs. We hope that the current version better separates our work from the work done in the past.**

Section 4.1, lines 236-245, Figure 5. As well as the west-east differences in variance explained by EOF1/EOF2, there are differences based on water depth, with EOF1 explaining the largest part of the variance in waters over 2000m deep (in the west) and EOF2 explaining more variance in waters less than 2000 m deep. What are the reasons for these differences? Are the warming/cooling signals associated with the SSH changes propagating in depth as well as space?

The likely reason is advection. We cannot say whether there is propagation in depth coordinate, but it appears that SSH anomalies first appear within the flows along the flanks of Reykjanes Ridge and along the East Greenland Current and then reach the deeper parts of the Irminger Basin and Labrador Sea. So, this might be a horizontal signal propagation due to eddies generated by those currents. In Section 5 (Discussion and Conclusions) of the revised manuscript, we have added the following sentences, which hopefully provide some discussion of the questions posed by the reviewer: "*It appears that while the overall propagation is westward, SSH anomalies associated with EOF1 and EOF2 first spread over the shallower areas in the east-northeast, including the currents along the eastern and western flanks of the Reykjanes Ridge and the East Greenland Current (EOF2; Figs. 2b and 5b), and then propagate towards the deeper parts of the Irminger Basin and Labrador Sea (EOF1; Figs. 2a and 5a). The horizontal transfer of signals from the currents to the interior basins may be carried by eddies generated by the boundary currents."*

Section 4.2, lines 263-274, Figure 7. I was a bit surprised at the percentage of SSH variance explained by  $SSH_{ST}$  in the WSPNA, as there appears to be a larger misfit between the black and blue lines in Figure 7b? For the period described in the text,  $SSH_{ST}$  decrease of 5 cm was part of an SSH decrease of nearly 8 cm. What explains these differences? Changing barotropic flows? Please include some discussion of the part of the SSH signal which is not captured in SSHST.

Indeed, the mismatch between SSH and  $SSH_{ST}$  in the WSPNA is larger, however, the correlation between these time series in the WSPNA is a little greater than in the ESPNA (0.99 vs 0.97). To reflect these points, we modified the text by replacing the explained variance information with

correlation and root-mean-squared differences between the time series as follows: "The time series of SSHST closely matches those of SSH, with the correlation between them above 0.95 and the root-mean-squared differences of 0.4 in the eastern SPNA and 0.8 cm in the western SPNA (compare black and blue curves in Fig. 7). This means that the variability of SSH in the SPNA is mostly steric in nature. The remaining difference between SSH and SSHST can be attributed to density changes at depths greater than 1000-m and to errors in data; the contribution of barotropic signals is expected to be small at the time scales considered."

Section 4.5. In the light of the previous sections on propagating signals, why were the periods here selected based separately on periods between maxima and minima of PC1 and PC2 (rather than phases of the propagating signal)? The first period (1994-2010) covers more than 2 periods of the propagating signal. And having chosen periods based on PC1 and PC2, why are they discussed in the order presented alternating PC1- and PC2-based periods? I think it would be useful to mark these periods on Figures 3 and 9.

Note that the modes have different spatial footprints, so that selecting time intervals characteristic for the main tendencies in each mode helps to better assess and visualize the role of buoyancy forcing in driving each individual mode, including the real and imaginary parts of CPC1. If we selected the time intervals based on the temporal phase of CEOF1, then EOF1 (real part of CEOF1) and EOF2 (imaginary part of CEOF2) would be mixed because of propagation. To add more clarity to the text, we have added the following sentence to the first paragraph of the revised manuscript: "*Because the two leading modes of variability have quite distinct spatial footprints, this allows to better assess and visualize the role of buoyancy forcing in driving each mode.*"

Regarding the order the periods are discussed, we think it is more logical to present them in chronological order. To alleviate a potential confusion, in the first paragraph of Section 4.5 in the revised version of the manuscript, we mention that the discussion of SSH changes during the selected time intervals follows a chronological order. Also, following the recommendation of the reviewer, we marked the time intervals in Fig. 3 by adding horizontal bars with diagonal stripes.

Section 4.5. This section discusses absolute SSH changes, rather than those reconstructed from the EOFs or CEOFs. How much difference does this make?

We believe it is more appropriate to compare the absolute SSH changes with the steric and those driven by surface fluxes. Note that the reconstructed SSH is a filtered version of SSH, and the two leading models of variability have different spatial footprints. For example, in 1994-2010, the SSH increase occurred mainly in the WSPNA, well depicted by the EOF1/PC1, while the SSH change in the ESPNA was small, because in this region SSH peaked earlier in 2004 and in 2010 it was already close to values in 1994. The difference between the absolute and reconstructed SSH in the SPNA is small, as suggested by Figs. 5 and 6.

Does the west-east propagating SSH contain less local surface flux signal and more advection, for example? Or is this not possible to determine?

It is an interesting question, but because the ocean is very dynamic and constantly exchanges properties with the atmosphere and ambient water masses, it is problematic to determine how much surface-generated and advective signal is contained in a propagating SSH anomaly with available observations.

Section 4.6. While opening with the 2011-2015 cold blob in the Iceland Basin as an example of an advective feature, the discussion doesn't really consider the source of that feature, just its subsequent westward, downstream advection. Can your analysis say anything about the upstream source of this feature?

The possible sources of the 'cold blob' are mentioned in the third paragraph of the introduction with relevant references. We have removed a reference to the 'cold blob' from the first paragraph of Section 4.6 and only mentioned that the observed cooling and freshening was caused by advection associated with the North Atlantic Current. Regarding the source of this feature, at the end of the fourth paragraph of Section 4.5 we have added the following sentence: "This agrees with a recent study by Holliday et al. (2020), who attributed the unprecedented freshening in the eastern SPNA in 2012-2016 to large scale changes in ocean circulation driven by atmospheric forcing."

It should be explained why velocities at 1000-dbar from Argo/altimetry are used when the cited method of Schmidt produces horizontal velocity estimates throughout the upper 1000m. It isn't clear to me, either from the methods or this results section, what different information is provided by the velocities and the eddy propagation velocities, and why both are used. I couldn't find the time-span over which these mean currents are calculated.

In Section 3.3 of the revised manuscript, we have provided more information on 1000-dbar velocities. We have also tried to stress that the 1000-dbar velocities and eddy propagation velocities complement each other.

Figures 10 and 11. These regressions are presumably come with an associated measure of significance? Perhaps consider including the arrows/colours only where they are significant?

In Figs. 10 and 11 of the revised manuscript, we have plotted arrows only at locations where regression coefficients are significant at 95% confidence.

**Technical corrections**

**3.1 EOF analysis**

This section appears to have some errors in the equations:

Eq. 1:  $EOF_j$  is just a function of position (not position and time). You could use bold for the position vector, **x**. Explain that N is the number of EOFs in the reconstruction and that  $SSH_R$  is reconstructed SSH

Thank you for noting this. We have made all suggested changes.

Eq. 2: I think the LHS of this equation is variance explained by the jth EOF, but the RHS is variance explained by the sum of the first N EOFs.

Correct! We have removed the subscript j from sigma on the LHS.

Eq. 3: LHS should make it clear it is the augmented complex SSH, perhaps call it  $SSH_C$  to differentiate it from SSH.

Thank you for catching this. We have changed the LHS to SSH\*.

Figures: Panel labels should be formatted as (a), (b), etc.

The labels have been changed.

Figure 3 caption refers to 'SSL' as opposed to SSH. I can't find reference to SSL in the text. Should SSL be SSH\_ST?

Thanks for noticing this. In an earlier version of the manuscript, SSH\_ST was called SSL. We have changed the caption accordingly.

Figure 11. I was very confused by the reversal of the blue-red color scale compared to everywhere else in the manuscript (to red negative, blue positive here), for a long time thinking the figure showed exactly the opposite thing to that being described. I understood (eventually) that this was to help the comparison with the SSH patterns in Fig 8. This reversal of the scale needs to be made much clearer or, preferably, return to the conventional use of the scale. I think readers can still easily compare figures 8 and 11.

We have decided to replace wind stress curl with sea level pressure (SLP) in Fig. 11. We believe that the revised figure is visually easier to read with respect to the shifting atmospheric circulation patterns and how they relate to the westward propagation of SSH anomalies.

Figures 10 and 11. I think these regression maps are anomalies, it isn't always clear when the full variable is meant, and when it is the anomaly.

Because these are regression maps, it does not matter whether the full variables or their anomalies are used for computation. The regression maps show the change in SLP/wind per 1 standard deviation change in PC1&2 and CPC1.

Figure 17. It is difficult to see the red, green and blue contours mentioned in the caption.

The figure has been revised.

**RC2:**

The subpolar North Atlantic is a key region for climate variability both over ocean and land. Using sea-surface heights and hydrography, this study by Volkov and colleagues adds new insights regarding the processes driving the interannual-to-decadal variability in the region. This is important in order not to generalize the processes, as typically done, dominating the characteristic time intervals during the altimetry period. The paper is well-written and the analysis are clear and straightforward but there are some elements that need improvement, and I am still missing the big picture of the presented results. I include some comments to help revising the paper before it can be recommended for publication.

We thank the reviewer for his/her time to review the manuscript and for the feedback. We have tried to address all reviewer's concerns by expanding the Discussion and Conclusion sections and by adding additional/clarifying sentences throughout the manuscript. Our answers to the reviewer's comments are shown below in blue font.

Comments

• The introduction presents the AMOC and North Atlantic gyres, and how they have varied over time, but the results in the paper do not connect back to these circulations. To give one example: what is the state of these circulations during the identified characteristic time intervals? A discussion section may be required to put the results in the paper in an AMOC and gyre context. The authors should also explore the possibility of using the observed AMOC time series to do so.

We thank the reviewer for this remark. In order to address it, we have changed the "Conclusions" sections to "Discussion and Conclusions", and we have added some discussion that hopefully links the results of the paper with what is said in the Introduction. We note, however, that the main objective of this paper is to revise the definition of the North Atlantic SSH tripole by accounting for signal propagation, as it applies to the subpolar North Atlantic. The detailed investigation of the tripole-related changes in circulation, the role of the AMOC, and air-sea interactions goes beyond the scope of this study and requires separate dedicated studies. The use of the AMOC time series in the SPNA is problematic, because the only AMOC time series, based on direct measurement by the OSNAP array, are still very short.

• The reader is also minimally provided with the implications of the results. Adding this may be easier after reconnecting the presented results to changes in the North Atlantic circulation (comment above).

In the Discussion and Conclusions section, we have added sentences and paragraphs regarding the implications of the results for the use of tripole and subpolar gyre index, as well as how the results link to inter-gyre exchange.

• The CEOF analysis applied to the SSH is technically straightforward to understand. But, what does this signal propagation tells us about the two-way coupling or communication between the two gyres during its evolution? And can this mainly be attributed to advection?

In the Discussion and Conclusions section of the revised manuscript, we have included the following paragraph: "It should be noted that due to geostrophy both the SSH and the general ocean circulation are linked, and both adjust to persistent atmospheric forcing. For example, an increase of SSH along the European coast starts when the negative (cyclonic) SLP anomaly is centered over the eastern coast of Greenland and the atmospheric circulation near the eastern boundary is likely to cause downwelling (phase 0° in Figs. 8 and 11). As the cyclonic SLP anomaly weakens and moves towards the Labrador Sea (phase 45° in Figs. 11), the subpolar gyre weakens and contracts and the positive SSH anomalies near the eastern boundary expand westward (phase 45° in Fig. 8). It has been reported that this situation can facilitate inter-gyre exchange (Häkkinen et al., 2011; Piecuch et al., 2017). Specifically, in response to a weakening of the subtropical high and subpolar low pressure centers, the subtropical and the subpolar gyres weaken, sea level decreases in the subtropical gyre and increases in the subpolar gyre, the subpolar front moves westward, and the eastern boundary region in the SPNA widens, entraining more warm and saline waters from the subtropical gyre. Consequently, positive SSH anomalies emerge first near the eastern boundary of the SPNA and then expand westward as the subpolar gyre continues to weaken (phases 45° to 180° in Fig. 8). The opposite occurs when the subtropical and the subpolar gyres strengthen (phases -135° to 0° in Fig. 8). As demonstrated by the  $CPC_1$  (Fig. 9), the local maximum SSH anomalies occurred in the eastern SPNA around 1996, 2004, and 2009, and they reached the western SPNA 1-2 years later. The most recent increase of SSH in the eastern SPNA since 2014 and in the western SPNA since 2016, that remains present in 2020, represents a recovery from an exceptional cooling and freshening that occurred in the SPNA in 2012-2016. This means that the recent conditions are favorable again for inter-gyre exchange."

We also conclude in the end that the observed interannual-to-decadal variability of SSH, including the westward propagation of SSH anomalies, is the result of a complex interplay between the local wind and surface buoyancy forcing, and the advection of properties by mean ocean currents.

• While the CEOF analysis applied to oceanic variables is "easy" to grasp, this is not the case for atmospheric variables and on long time scales (there is no such predictability in the atmosphere, is there?). Because of this, I am really struggling in interepting the results presented in Fig. 11.

In the revised version of the manuscript, we have replaced wind stress curl in Fig. 11 with sea level pressure. We believe that the new figure is visually more comprehensive with respect to the tripole-related shifts of wind forcing patterns. Our observation is that the westward propagation of SSH anomalies corresponds to the shifts of wind forcing patterns in the same direction. Given this observation, it is possible that the observed propagation is simply an oceanic response to persistent low-frequency NAO-like wind forcing that is not stationary (as described by the conventional NAO index). However, in the presence of the mean cyclonic circulation in the SPNA, advection also plays an important role, as also demonstrated by (mainly) halosteric SSH anomalies in Figs. 12-15. Furthermore, persistent wind forcing leads to baroclinic adjustment with associated changes in heat and freshwater contents and in geostrophic circulation. We agree that our study does not provide a

conclusive answer to what process is more important in driving the tripole-related changes in the SPNA: wind forcing, buoyancy fluxes, or advection. This question is beyond the scope of this paper and needs to be addressed in future studies. In the last paragraph of the Discussion and Conclusions section we note the following: "Overall, we conclude that the observed interannual-to-decadal variability of SSH, including the westward propagation of SSH anomalies, is the result of a complex interplay between the local wind and surface buoyancy forcing, and the advection of properties by mean ocean currents. The relative contribution of each forcing term to the variability is space and time dependent and, therefore, impossible to assess with available observations."

**Minor comments**

• Why does EOF2 show a strong signal on the shelves?

This is an interesting question, similar to the one posed by reviewer 1. The EOF2 shows a strong signal over shallower areas, which are present mainly in the eastern and northeastern SPNA. The western SPNA is largely deep. The SSH anomalies depicted by EOF2 first emerge near the eastern boundary, which is probably due to the associated atmospheric circulation pattern as well as to the fact that the subpolar gyre contracts or expands allowing more or less subtropical waters to enter the SPNA. We believe that the likely influence of advection is reflected by the bands of high explained variance along the flanks of Reykjanes Ridge and along the East Greenland Current (Fig. 5b). In the Discussion and Conclusions section of the revised manuscript, we have added the following sentences: "It appears that while the overall propagation is westward, SSH anomalies associated with EOF1 and EOF2 first spread over the shallower areas in the east-northeast, including the currents along the eastern and western flanks of the Reykjanes Ridge and the East Greenland Current (EOF2; Figs. 2b and 5b), and then propagate towards the deeper parts of the Irminger Basin and Labrador Sea (EOF1; Figs. 2a and 5a). This suggests that advection by the mean currents in the SPNA is an important factor to consider and that the transfer of signals from the currents to the interior parts of the basins may be due to eddies."

• 260: How different is the variability of the SSH between the Iceland and Rockall Basins as compared to that of the Irminger Sea?

The SSH time series in the regions are highly correlated at the time scales considered. Therefore, we have removed the paragraph mentioning that the regions are dynamically distinct.

• 315: What is the reason for the 10-month lag?

We have added the following sentences to the first paragraph of Section 4.4: "The lag probably indicates the oceanic adjustment time to a variable wind forcing. Regression of SLP and 10-m winds on the PC1 displays a familiar NAO dipole pattern with a cyclonic (negative) anomaly in the subtropical high and an anticyclonic (positive) anomaly in the subpolar low SLP centers (Fig. 10a). The weaker/stronger subtropical high and subpolar low associated with weaker/stronger westerly winds in the midlatitude North Atlantic lead to lower/higher sea levels in the subtropical North Atlantic and higher/lower sea levels in the SPNA."

• 325: Can the authors quantify the Ekman-induced SSH anomalies?

This would require the estimation of the reduced gravity, which is latitude dependent, and would make the manuscript even longer than it is now. We prefer to keep the main focus on the observed propagation and limit the analysis of wind forcing to establish a statistical relationship with the tripole-related changes of SSH. The mechanistic details of the relationship between the shifting wind forcing patterns and the tripole evolution in the SPNA are left for a future study.

• Figure 11: reversing the colormap is confusing.

We have replaced the wind stress curl in Fig. 11 with sea level pressure, so the colormap is not reversed anymore.

• Figure 12: please add more details to the caption.

We are not sure what additional details are meant, but we have modified the caption a little bit by listing abbreviations used in the figure panels and by indicating that steric sea level was estimated from EN4 data and the buoyancy fluxes are provided by the ERA5 reanalysis.

• Please assess and add significance to the regression figures throughout.

In Figs. 10 and 11 of the revised manuscript, we have plotted arrows only at locations where regression coefficients are significant at 95% confidence.

• Please improve the text whenever possible. It is hard to follow at multiple places in the results, especially when describing the results in the context of previous work. A discussion section would have helped to avoid this.

We have done a substantial revision of the Discussion and Conclusions sections as well as we have added clarifying sentences throughout the manuscript. We hope that the revised version has been improved and become more comprehensive.

---

## Author Response (AR2)

**Reviewer 2**

I think the authors have done a good job in revising their work and replying to most of the comments by both reviewers.

We thank the reviewer for the time spent to evaluate our manuscript, and we are happy that the revised version of the manuscript addressed most of the previous comments. We are also grateful for additional remarks, which we have tried to clarify both in this reply and in the manuscript. Our answers to reviewer's comments are given in blue font. In the revised manuscript, all new changes introduced are highlighted by dark red.

1) I am still struggling with the CEOF analysis applied to the SLP. With the analysis presented in Fig. 11, you are still expecting that the atmosphere is the main driver of the SSH anomaly as it propagates around the basin. An SSH anomaly will do so without any influence of the atmosphere as soon as it is introduced to the eastern SPNA. If the authors insist in keeping this analysis, the rational should be explicitly laid out.

We understand the reviewer's concern. In fact, the reviewer raises one of the questions, which lies beyond the scope of this study and requires further dedicated investigation. We should first clarify that we did not apply the CEOF analysis to the SLP. Fig. 11 shows regression maps of SLP and wind velocity on CPC1 computed for SSH anomalies. So, by comparing Figs. 8 and 11, we report on the correlation between the tripole-related SSH changes and shifting SLP/wind forcing patterns. We agree with the reviewer that an SSH anomaly generated in the eastern SPNA should be carried towards the Labrador Sea by the mean ocean circulation even without any additional wind forcing, albeit we do not know how quickly it would dissipate. On the other hand, we believe that it is important to report on the observed correlation between the propagating SSH anomalies and shifting SLP patterns, as it suggests that wind forcing is a possible driver as well. Physical mechanisms of the air-sea coupling in the SPNA are beyond the scope of this study, but we hope that the results of our investigation will motivate further research. To address the reviewer's concern we have modified the last three sentences of the manuscript as follows:

"While we identify shifting wind forcing patterns and the mean ocean circulation as possible drivers for the observed westward propagation of SSH anomalies in the SPNA on the interannual-to-decadal time scales, it remains unexplored which process is more important. Further research is needed to understand the mechanisms of air-sea coupling in the SPNA, including potential oceanic feedback on the atmospheric circulation. Finally, it is of particular interest to explore how the tripole variability is related to inter-gyre exchange and to the AMOC."

2) The new paragraph in the last section (Discussion and Conclusions) on gyre strength and subpolar front changes seems speculative. The authors need to be careful with these connections and show evidence of these.

This paragraph was added to address the reviewer's question in the first review: What does this signal propagation tell us about the two-way coupling or communication

between the two gyres during its evolution? To address the concern with this addition we slightly edited the text in the paragraph trying to improve its message (lines 539-554). This should make it clear to readers that we discuss observations and figures presented in the manuscript in the context of a previously suggested hypothesis on inter-gyre exchange (Hakkinen et al., 2011; Piecuch et al., 2017). With the revisions of this paragraph we believe that it is well suitable for the discussion section.

3) Please add relevant references to the role of eddies, page 18, line 1 "...to the interior basins may be carried by eddies generated by the boundary currents."

We thank the reviewer for pointing this out, and we have added related references in lines 548-549 (Fan et al., 2013; de Jong et al., 2014).